# An interactive murine single-cell atlas of the lung responses to radiation injury

Sandra Curras-Alonso [1,2], Juliette Soulier [1,2], Thomas Defard [3,4,5,6], Christian Weber[6], Sophie Heinrich [2], Hugo Laporte[1,2], Sophie Leboucher [7], Sonia Lameiras[8], Marie Dutreix [2], Vincent Favaudon[2], Florian Massip [3,4,5], Thomas Walter[3,4,5], Florian Mueller [6], José-Arturo Londoño-Vallejo [1,2] ✉ & Charles Fouillade [2] ✉

Radiation Induced Lung Injury (RILI) is one of the main limiting factors of thorax irradiation, which can induce acute pneumonitis as well as pulmonary fibrosis, the latter being a life-threatening condition. The order of cellular and molecular events in the progression towards fibrosis is key to the physio-pathogenesis of the disease, yet their coordination in space and time remains largely unexplored. Here, we present an interactive murine single cell atlas of the lung response to irradiation, generated from C57BL6/J female mice. This tool opens the door for exploration of the spatio-temporal dynamics of the mechanisms that lead to radiation-induced pulmonary fibrosis. It depicts with unprecedented detail cell type-specific radiation-induced responses associated with either lung regeneration or the failure thereof. A better understanding of the mechanisms leading to lung fibrosis will help finding new therapeutic options that could improve patients' quality of life.

Radiation therapy is one of the main therapeutic options used to treat thoracic cancers. Nevertheless, the lung is a sensitive organ to ionizing radiation (IR) which makes it the main dose-limiting organ in the thorax[1,2]. Radiation-induced lung injury (RILI) results in both early and late molecular and cellular toxicities, some of them irreversible, impacting the quality of life of patients. IR induces DNA damage and oxidative stress, followed by inflammation and tissue reorganization, which may be resolved into regeneration and regain of organ function. However, depending on the total dose as well as patient sensitivity, radio-induced pneumonitis may evolve toward radiation-induced pulmonary fibrosis (RIPF), characterized by fibroblast and myofibroblast proliferation, excessive extracellular matrix (ECM) deposition, bronchiolization, and honeycomb cyst formation, leading to disruption of gas exchange and progressive organ failure[3,4].

The mechanisms that determine success or failure of tissue regeneration after irradiation are not well understood. At the cellular level, pro-fibrogenic doses of irradiation trigger apoptosis, senescence, cytokine secretion, as well as cell transitions likely affecting multiple cell compartments. Single-cell technologies have provided powerful tools to untangle the complex cellular heterogeneity of organs such as the lung, an exceptional challenge in the respiratory field. In the past years, the Lung Biological Network of the Human Cell Atlas (HCA) has emerged with the objective of establishing a complete atlas of the healthy human lung, collecting data on molecular phenotypes of different cell types, cell transitions as well as their location all along the airways. In addition, LungMAP has gathered scRNA-seq datasets to provide a collaborative and open-access comprehensive molecular atlas of the healthy developing lung[5]. In parallel, other consortia have gathered efforts to build the Lung Disease Cell Atlas

[1]Institut Curie, CNRS UMR 3244, Sorbonne Universite, PSL University, 75005 Paris, France. [2]Institut Curie, Inserm U1021-CNRS UMR 3347, University Paris-Saclay, PSL University, Centre Universitaire, 91405 Orsay Cedex, France. [3]Centre for Computational Biology (CBIO), Mines Paris, PSL University, 75006 Paris, France. [4]Institut Curie, PSL University, 75005 Paris, France. [5]INSERM, U900, 75005 Paris, France. [6]Imaging and Modeling Unit, Institut Pasteur, Université Paris Cité, Paris, France. [7]Institut Curie, CNRS UMR 3348, University Paris-Saclay, PSL University, Centre Universitaire, Orsay, France. [8]Institut Curie Genomics of Excellence (ICGex) Platform, Paris, France. ✉e-mail: arturo.londono@curie.fr; charles.fouillade@curie.fr

(e.g., the IPF cell atlas[6,7], the COVID cell atlas[8], the COPD cell atlas[9]) to better describe lung cell and tissue responses associated with disease. Such efforts have allowed to build an integrated cell atlas of the human lung in health and disease[10]. Nevertheless, to date, no single cell-based efforts have been aimed at studying RIPF. Understandably, progresses in that direction face major obstacles, in particular, due to very limited access to specimens from irradiated human lungs. In this context, well-characterized mouse models of RIPF, which recapitulate major features of the disease progression in the human (namely, an early period of inflammation, irreversibly followed by fibrosis and death) carry high information value.

The purpose of this work is to provide a whole organ single-cell atlas spanning the evolution over time towards pulmonary fibrosis, from the early response and the inflammatory phase to the end-point fibrotic process. This communication illustrates the potential of the information provided by this atlas by focusing on representative cell types from the different cell compartments: epithelial (AT2 cells), mesenchymal (fibroblasts and myofibroblasts), myeloid (AMs and IMs), and endothelial (aCap and gCap) cells. We have combined single-cell RNA sequencing (scRNA-seq) with single-molecule fluorescence in situ hybridization (smFISH) to confirm specific findings and have analyzed using computational tools the dynamics in cell-cell communications to pinpoint biologically meaningful interactions that might be involved in the evolution of RIPF.

## Results
### Cellular composition dynamics after lung irradiation
To study the impact of radiation therapy in the lung, we performed scRNA-seq using the 10x Chromium Controller V3 technology of dissociated lungs from non-irradiated mice (control) and from mice 1, 2, 3, 4, and 5 months after fibrogenic (17 Gy) and non-fibrogenic (10 Gy) doses of IR[11]. We used 5 age-matched, non-IR mice as control, 5 mice after 10 Gy thorax IR (one mouse per time point from 1 to 5 months) and 10 mice after 17 Gy thorax IR (two mice per time point from 1 to 5 months) (Fig. 1a). Each mouse lung was enzymatically and mechanically dissociated into a single cell suspension that was then loaded into the 10x microfluidic system.

A total of 102,869 cells were obtained, from which 22,378 belonged to the healthy mice, 26,360 to the $IR_{10Gy}$ mice and 54,131 to the $IR_{17Gy}$ mice. UMAP visualization after merging the total dataset or per condition displayed similar distributions of cells into clusters, both in the control and in the two IR conditions over time, which indicated the absence of obvious batch effects due to sample processing or sequencing, thus underlining the reproducibility of the single cell experimental procedure and analysis (Fig. 1b, Supplementary Fig. 1a–c).

We used cell type-specific markers from recently published scRNA-seq datasets[12–14] to annotate major populations. This resulted in the identification of 18 main cell types (Fig. 1b, c): non-immune cells, which comprise 4 epithelial cell populations (AT2 cells, AT1 cells, club cells, and ciliated cells), 3 mesenchymal cell clusters (fibroblasts, smooth muscle cells -SMC- and mesotheliocytes), 1 endothelial cell -EC- cluster and several clusters of immune cells, which include 7 myeloid compartments (monocytes, alveolar macrophages -AM-, interstitial macrophages -IM-, dendritic cells -DC-, plasmacytoid dendritic cells -pDC-, neutrophils and basophils) and 3 lymphoid compartments (T cells, natural killer -NK- cells and B cells). We also identified subpopulations of proliferating DC, AM, and T cells.

As a first approach to the study of RIPF, we sought to detect changes in cell proportions affecting the different compartments after irradiation. These results, illustrated in Fig. 1 and Supplementary Fig. 1, pointed to changes in the proportions of immune cells, which tended to increase after 10 Gy or 17 Gy irradiation, e.g., alveolar macrophages (Supplementary Fig. 1d), and of epithelial cells, which tended to progressively decrease, in particular after 17 Gy irradiation (Fig. 1d;

Supplementary Fig. 1e). Similar observations were made concerning the endothelial compartment (Fig. 1d; Supplementary Fig. 1f). However interesting, extreme caution should be exerted when interpreting such results as they can be easily biased due, for instance, to the relative efficiency in tissue dissociation (in particular under fibrotic conditions). Also, given the limits imposed by the technology in the number of cells analyzed per sample, changes in the numbers of one cell compartment will necessarily affect the calculated proportions in other compartments. Taking these considerations into account, in the following sections we coupled scRNA-seq analyses with smFISH experiments to obtain independent confirmation of changes in cell proportions affecting AT2 cells, macrophages, endothelial cells, and fibroblasts.

### Transcriptomic dynamics in AT2 cells point to transdifferentiation in response to fibrogenic irradiation
AT2 cells play an important role in the lung as they secrete surfactants that maintain surface tension and prevent alveolus collapse. They are also key elements in lung homeostasis because of their stem cell capacity and ability to differentiate into AT1 cells, the latter being responsible for air exchange. In agreement with the detected decrease in the proportion of epithelial cells, the total number of AT2 cells per sample was decreased in all samples after IR. AT2 cells represented 12,6% (SD 3,92%) in the control mice. This proportion decreases after $IR_{10Gy}$ but it is more progressive and pronounced after $IR_{17Gy}$ (Fig. 2a, Supplementary Fig. 2a). We validated this observation in smFISH experiments where NI, $IR_{10Gy}$, and $IR_{17Gy}$ lung tissue sections were probed against *Lamp3*, a robust and specific marker for AT2 cells (Supplementary Fig. 2b). As shown in (Fig. 2b, c, Supplementary Fig. 2c), there is indeed a significant decrease in the AT2 cell proportion in the 17 Gy condition after 5 months. Interestingly, the volume of the AT2 cells, which was estimated based on the distribution of *Lamp3* smFISH signal in the cytoplasm of these cells (see Methods), showed a significant increase in this particular condition, while it was not altered after $IR5M_{10Gy}$ (Fig. 2c), suggesting possible significant changes in the genetic program exclusively associated with pro-fibrogenic IR.

The distribution of the irradiated samples within the AT2 cells cluster did not entirely overlap with the control samples (Supplementary Fig. 2a), suggesting IR-induced transcriptional changes. Differentially expressed genes (DEG) analysis revealed qualitative and quantitative differences in the transcriptional response to $IR_{17Gy}$ and to $IR_{10Gy}$ (Fig. 2d). After IR17Gy, the number of upregulated genes increased steadily up to 5M. In comparison, the response to $IR_{10Gy}$ was a transient one, reaching a peak after 3 months post-IR and decreasing afterwards (Fig. 2d). Amongst the genes differentially expressed by irradiated AT2 cells there were genes typically involved in Epithelial-Mesenchymal transition (EMT) (e.g., *Serpine2, Tgfbr2, Col6a2, Col4a1, Col4a2, Colgalt1, Egfl6*)[15]. To better characterize EMT dynamics in AT2 cells after IR, we selected 31 EMT-associated genes that were significantly upregulated after $IR5M_{17Gy}$ and followed their expression from 1 to 5 months after 10 and 17 Gy IR (Supplementary Fig. 2h). Interestingly, while all these genes were only slightly and transiently upregulated after $IR_{10Gy}$, they only showed strong upregulation at 4 and 5 months after $IR_{17Gy}$ (Supplementary Fig. 2h–j), that is, during the fibrogenic phase. Whether EMT in AT2 cells directly contributes to RIPF or constitutes a secondary response to the pro-fibrotic micro-environment remains to be determined.

Another look at the list of deregulated genes after a pro-fibrogenic dose revealed a number of genes known to be specific markers of AT1 cells (e.g., *Akap5, Aqp5*). This observation suggested that, similar to what has been described in response to other types of injuries[16], AT2 cells may undergo transdifferentiation towards AT1 in response to IR. To further explore this possibility, we analyzed the expression of the 30 genes composing the AT2 > AT1 transdifferentiation signature. Strikingly, 1 M after 17 Gy irradiation, 30% of these genes are

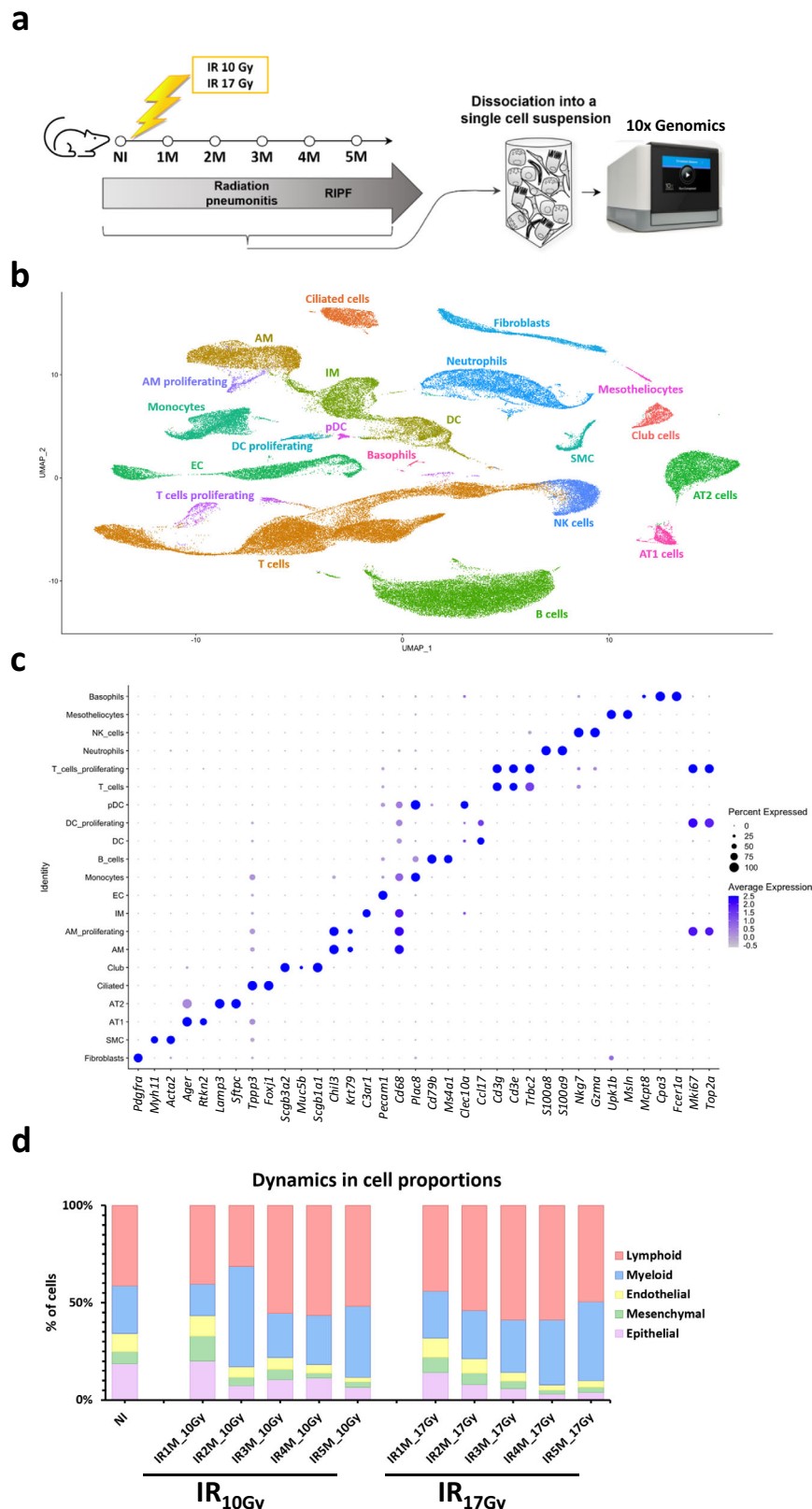

**Fig. 1 | Radiation-induced lung injury cell atlas. a** Scheme of the experimental set up. Mice are irradiated with a dose of 10 Gy or 17 Gy and monitored by CT-scan to follow the development of pulmonary fibrosis. Mice are sacrificed at 1, 2, 3, 4, and 5 months after IR, and lungs are enzymatically and mechanically dissociated into a single cell suspension before being loaded in the 10x Chromium Controller System (image provided by 10x Genomics). **b** UMAP visualization of 102,869 cells from 20 different samples (5 NI; 5 IR$_{10Gy}$, one per time point; 10 IR$_{17Gy}$, two per time point) annotated by cell type. **c** Dot plot of the expression of the markers used for cell type identification. **d** Dynamics in cell proportions of the endothelial, mesenchymal, epithelial, lymphoid, and myeloid cells across the NI and IR conditions at the different time points and doses.

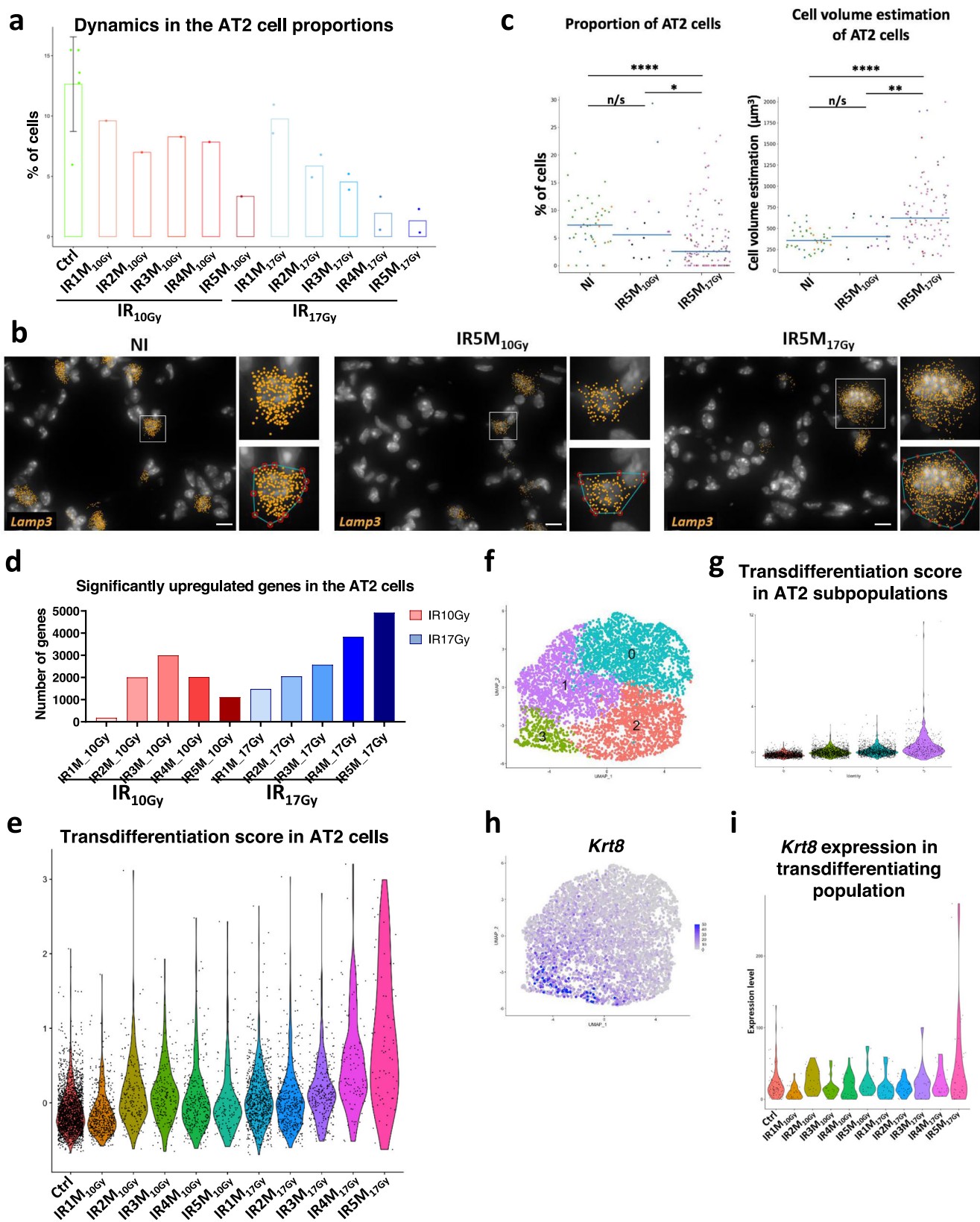

**a** Dynamics in the AT2 cell proportions

**c** Proportion of AT2 cells        Cell volume estimation of AT2 cells

**b** NI            IR5M₁₀Gy            IR5M₁₇Gy

**d** Significantly upregulated genes in the AT2 cells

**f**

**g** Transdifferentiation score in AT2 subpopulations

**e** Transdifferentiation score in AT2 cells

**h** *Krt8*

**i** *Krt8* expression in transdifferentiating population

upregulated, a proportion that increases progressively over the following months, reaching 63,3% at 5 M after IR₁₇Gy (Supplementary Fig. 2d). This observation lends support to the assumption that a process of AT2 > AT1 transdifferentiation occurs progressively after IR₁₇Gy reaching its strongest expression when fibrosis is already declared (Fig. 2e, Supplementary Fig. 2d). In contrast, when a

non-fibrogenic dose is given (IR₁₀Gy), such transdifferentiation appears to be limited both in strength (26.7% of the genes) and time, reaching its maximum at 3 months after IR₁₀Gy (Fig. 2e, Supplementary Fig. 2d).

More precisely, we observed a particular subpopulation of AT2 cells that showed the highest transdifferentiation score, the cluster 3 (Fig. 2f,g). This cluster was characterized by the expression of *Krt8*

**Fig. 2 | Cellular and molecular changes in the AT2 cells during RILI reveal a transdifferentiation profile after fibrogenic doses of IR. a** Dynamics in the proportion of the AT2 cells in the NI ($n = 5$) and at the different time points after IR$_{10Gy}$ ($n = 1$) and IR$_{17Gy}$ ($n = 2$). Error bar refers to the standard deviation of the data. **b** Automatic *Lamp3* mRNA (orange) detection with Big-FISH in NI, IR5M$_{10Gy}$, and IR5M$_{17Gy}$ lung tissue sections. Inset top panel shows an AT2 cell; inset bottom panel shows the convex hull of a cluster of mRNA spots. Scale bars, 10 μm. **c** Quantification and cell volume estimation of the Lamp3+ cells in NI, IR5M$_{10Gy}$, and IR5M$_{17Gy}$ lung tissue sections. To compare two groups, the *P* value was computed with the Mann–Whitney–Wilcoxon test (two-sided test) from scipy (n/s, adjusted *p* value >0.05; *, adjusted *p* value <0.05; **, adjusted *p* value <0.01; ***, adjusted *p* value <0.001; ****, adjusted *p* value <0.0001). Each dot represents one analyzed image. Each color per time point represents a different biological replicate (NI $n = 3$; IR5M$_{10Gy}$ $n = 3$; IR5M$_{17Gy}$ $n = 5$). **d** Dynamics in the significantly upregulated genes in the AT2 cells compared to the NI samples at the different time points after IR$_{10Gy}$ and IR$_{17Gy}$. **e** Violin plot showing the single cell score calculated based on the transdifferentiation expressed genes in the AT2 cells. **f** UMAP visualization of the different AT2 cell subpopulations. **g** Violin plot showing the single cell score calculated based on the transdifferentiation expressed genes in the different AT2 cell subpopulations. **h** UMAP visualization of the expression of *Krt8*. **i** Violin plot of *Krt8* expression in the AT2 cells cluster three in the NI samples and at the different time points after IR$_{10Gy}$ and IR$_{17Gy}$.

(Fig. 2h), which was mainly upregulated 5 months after 17 Gy irradiation (Fig. 2i). Previous studies have demonstrated the existence of an alveolar epithelial *Krt8* + transitional stem cell state that derives from activated AT2 cells and differentiates into AT1 cells[17], suggesting that the cluster 3 may transdifferentiate towards AT1 cells. To address this hypothesis, we performed a trajectory analysis using AT2 and AT1 cells which shows the connection between these two epithelial populations of the alveolus (Supplementary Fig. 2e). Moreover, the pseudotime analysis using the transdifferentiation-associated genes to order the cells suggests that AT2 cells differentiate towards AT1 cells (Supplementary Fig. 2f). To identify specific regulatory factors, we ran SCENIC on the AT2 subset (Supplementary Fig. 2g). This regulatory network analysis showed the activation of four transcription factors at 4 and 5 months post irradiation: *Stat1*, *Stat3*, *Irf7*, and *Xbp1*. Interestingly, *Stat1* and *Stat3* have previously been shown to be activated during the development of idiopathic pulmonary fibrosis[18].

Overall, these results indicate that there is a progressive loss of AT2 cells only after a fibrogenic irradiation dose and that this loss is associated with progressive and profound changes in the transcriptome landscape pointing to physiopathogenic transdifferentiation processes.

## Fibrogenic irradiation triggers a strong ECM genes response in fibroblasts

Lung fibroblasts are crucial for maintaining the integrity of the alveolar structure and play key roles in the response to injury through proliferation and remodeling of surrounding tissue. Specific analysis of the fibroblast compartment from all lung samples allowed us to distinguish 3 different sub-populations (Fig. 3a): two different matrix fibroblasts, one *Col13a1*-positive (*Col13a1*+, *Tcf21*+) and one *Col14a1*-positive (*Col14a1*+, *Pi16*+, *Meg3*+)[13], and one sub-population of myofibroblasts (*Hhip*+, *Cdh11*+, *Pdgfrb*+)[13,19,20] (Supplementary Fig. 3a). Surprisingly, most fibroblasts obtained 5 months after fibrogenic irradiation clustered with the latter (Fig. 3a, b) and represented over 80% of all fibroblasts in these samples (Fig. 3c). The analysis of the dynamics affecting fibroblasts subpopulations after irradiation indicated that the proportion of myofibroblasts increased dramatically 4 and 5 months after 17 Gy irradiation in detriment of both types of matrix fibroblasts (Fig. 3c). This observation was unexpected since lungs appeared to be quite fibrotic when processed, especially after 5 M post-17Gy irradiation. To corroborate this information, we performed smFISH using a probe to detect *Pdgfra*, a general marker for fibroblasts (Supplementary Fig. 3b). As shown in Fig. 3d, e, the number of fibroblasts detected per field of view is significantly higher 5 months after irradiation which is in striking contrast to the numbers inferred from the scRNA-seq analyses. This observation suggests a bias in the cell composition after the dissociation of injured lungs, in particular under fibrogenic conditions. On the other hand, a smFISH analysis using probes to detect *Hhip*, a specific marker for myofibroblasts, also indicated a higher proportion of these cells detected per field of view 5 months after fibrogenic irradiation (Fig. 3d, e). Finally, a small number of fibroblasts appeared to co-express both *Pdgfra* and

*Hhip*, and this number appeared to increase 5 months after irradiation (0.4% in NI to 2.8% IR5M$_{10Gy}$ and to 4.2% IR5M$_{17Gy}$).

Despite the detected bias in cell composition at late time points revealed by smFISH, we attempted a fibroblast DEG analysis between samples. This analysis indicates that the number of differentially upregulated genes increases steadily after 17 Gy irradiation (Supplementary Fig. 3c). As predicted, the transcriptomic response of fibroblasts to irradiation comprises genes known to impact ECM deposition, which were significantly upregulated in all three compartments (Fig. 3f, Supplementary Fig. 3d). None of these ECM genes were significantly upregulated 4 and 5 M after IR$_{10Gy}$, supporting the notion that this is part of the toxic response of fibroblasts specifically associated to lung fibrosis.

## Specific macrophage compartments display either proinflammatory or profibrotic profiles after IR17Gy

Analysis of the 11,678 macrophages from NI, IR$_{10Gy}$ and IR$_{17Gy}$ samples at different time points identified two main macrophages populations (Fig. 4a, b): alveolar macrophages (AM), characterized by the expression of *Chil3* and *Plet1* (Supplementary Fig. 4a) and interstitial macrophages (IM), which shows high levels of expression of *C1qa* and *C1qb* (Supplementary Fig. 4a). Further analysis showed that IMs could be subdivided in three different subsets; the first subset is characterized by the expression of *Folr2*, *Ccl8* and *Cd163* (called here IM_C1), the second subset expresses *H2-DMa*, *Zmynd15* or *Cd63* (IM_C2), and the third subset shows high levels of expression of *Ccr2* and *S100a4* (IM_C3) (Fig. 4a, Supplementary Fig. 4a). Similarly, AMs were distributed into 2 different subpopulations: one characterized by the expression of *Krt19*, *Fabp1* and *Krt79* (here called AM_C1) and the other one characterized by high levels of *Chil3*, *Wfdc21* and *Ctsd* (AM_C2) (Fig. 4a, Supplementary Fig. 4a). Macrophages from all samples were not homogenously distributed in these subpopulations. For instance, the AM_C2 subpopulation was particularly enriched in AMs obtained 5 months after 17 Gy irradiation (Fig. 4b). This alveolar macrophage subpopulation appeared almost exclusively after irradiation and was already strongly reinforced at 4 months after 17 Gy (Fig. 4c) reaching 72-78% of the total AMs. The changes that affected the proportions of IMs after irradiation were more subtle, although a relative increase in IM_C3 is noticeable 4 and 5 months after 17 Gy irradiation (from 11.4% in the NI to 38.6% in the IR5M) (Fig. 4c). Furthermore, an increase in the numbers of both AMs and IMs was confirmed in the lungs by smFISH 5 months post-17Gy while this is not the case after 10 Gy irradiation (Fig. 4d, e, Supplementary Fig. 4c).

DEG analysis of IMs showed an upregulation of markers known to define the macrophage M1 activated state (e.g., *Ccr2*, *Stat1*)[21–23], as well as high levels of proinflammatory cytokines and chemokines (e.g., *Nfkb1*) mainly after IR$_{17Gy}$, and to a lesser extent after IR$_{10Gy}$ (Fig. 4f). Moreover, the upregulation of these inflammatory genes seemed to specifically occur within the IM_C3 cluster (Fig. 4f, h). These results suggest the emergence of a pro-inflammatory M1 population of IM during the fibrogenic phase at 4 and 5 M after IR$_{17Gy}$ (Fig. 4f). Similar DEG analysis of the AMs showed the upregulation of genes related to a profibrotic response (e.g., *Lpl*), as well as other markers known to be

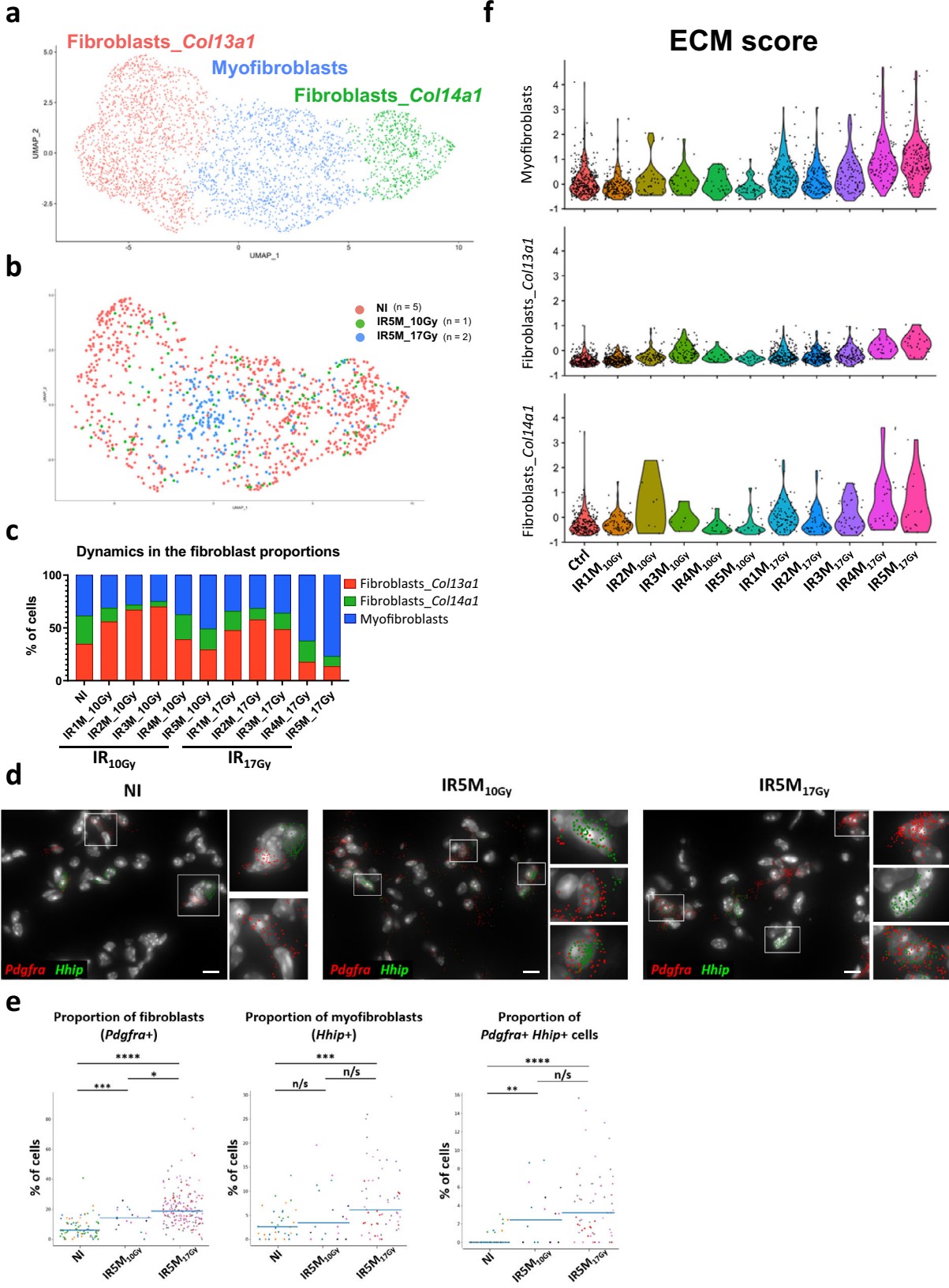

characteristic of the macrophage M2 activated phenotype (e.g., *Tgm2*)[21], along with Th2 cytokines receptors (e.g., *Il4ra*) (Fig. 4g, i) and genes that have been related to foam cells (e.g., *Cd63, Nr1h3, Abcg1*) (Supplementary Fig. 4d). These profibrotic markers were over-expressed in the AM_C2 subpopulation, which was specifically enri-ched after IR, and became predominant 4-5 months after 17 Gy

irradiation, suggesting particular toxic role conditions under pro-fibrotic conditions.

Strikingly, this specific transcriptional change of AM from AM_C1 to AM_C2 is accompanied by morphological changes detected in smFISH: the estimated volume of the AMs is increased 5 months after $IR_{17Gy}$ (from 459 $\mu m^3$ to 925 $\mu m^3$), while it remains constant after

**Fig. 3 | Myofibroblasts contribute to the ECM deposition after IR₁₇Gy. a** UMAP visualization of 3488 cells from the different fibroblast subpopulations annotated by cell type. **b** UMAP visualization of NI (*n* = 5), IR5M₁₀Gy (*n* = 1) and IR5M₁₇Gy (*n* = 2) fibroblasts annotated by time point. **c** Dynamics in the proportion of the fibroblast subpopulations at the different time points after IR₁₀Gy and IR₁₇Gy. **d** Automatic *Pdgfra* (red) and *Hhip* (green) mRNA detection with Big-FISH in NI, IR5M₁₀Gy, and IR5M₁₇Gy lung tissue sections. Scale bars, 10 μm. **e** Quantification of the *Pdgfra*+, *Hhip*+ and *Pdgfra*+ *Hhip*+ cells in the NI, IR5M₁₀Gy and IR5M₁₇Gy lung tissue sections.

To compare two groups, the *P* value was computed with the Mann–Whitney–Wilcoxon test (two-sided test) from scipy (n/s, adjusted *p* value >0.05; *, adjusted *p* value <0.05; **, adjusted *p* value <0.01; ***, adjusted *p* value <0.001; ****, adjusted *p* value <0.0001). Each dot represents one analyzed image. Each color per time point represents a different biological replicate (NI *n* = 3; IR5M₁₀Gy *n* = 3; IR5M₁₇Gy *n* = 5). **f** Violin plot showing the single cell score calculated based on the ECM expressed genes in the myofibroblasts, fibroblasts *Col13a1,* and fibroblasts *Col14a1*.

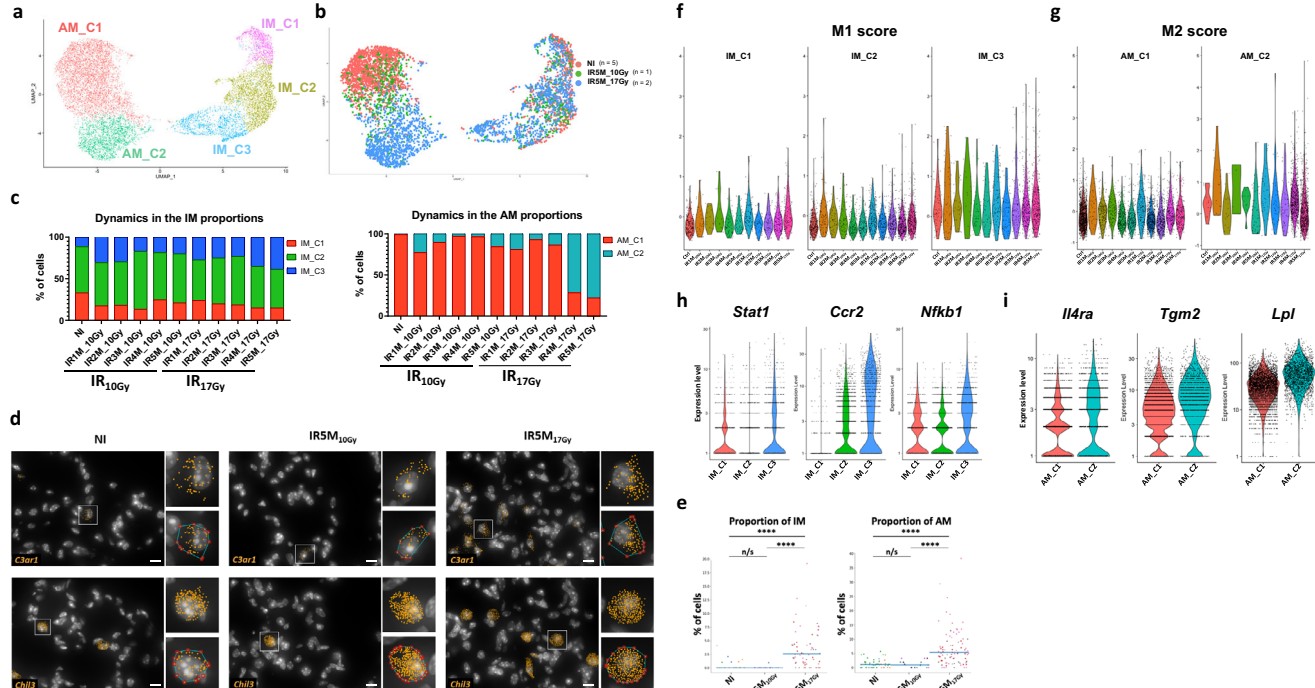

**Fig. 4 | Proinflammatory and profibrotic profile of alveolar and interstitial macrophages after fibrogenic doses of IR. a** UMAP visualization of 11,678 cells from the different IM and AM subpopulations annotated by cell type. **b** UMAP visualization of NI (*n* = 5), IR5M₁₀Gy (*n* = 1) and IR5M₁₇Gy (*n* = 2) IM and AM annotated by time point. **c** Dynamics in the proportion of the IM and AM subpopulations at the different time points after IR₁₀Gy and IR₁₇Gy. **d** Automatic *C3ar1* (orange) and *Chil3* (orange) mRNA detection with Big-FISH in NI, IR5M₁₀Gy and IR5M₁₇Gy lung tissue sections. Scale bars, 10 μm. **e** Quantification of the *C3ar1*+ and *Chil3*+ cells in the NI, IR5M₁₀Gy, and IR5M₁₇Gy lung tissue sections. To compare two groups, the *P* value was computed with the Mann–Whitney–Wilcoxon test (two-sided test) from scipy

(n/s, adjusted *p* value >0.05; *, adjusted *p* value <0.05; **, adjusted *p* value <0.01; ***, adjusted *p* value <0.001; ****, adjusted *p* value <0.0001). Each dot represents one analyzed image. Each color per time point represents a different biological replicate (NI *n* = 3; IR5M₁₀Gy *n* = 3; IR5M₁₇Gy *n* = 5). **f** Violin plot showing the single cell score calculated based on the M1 signature in the different IM subpopulations. **g** Violin plot showing the single cell score calculated based on the M2 signature in the different AM subpopulations. **h** Violin plots of M1 genes expression in the different IM subpopulations. **i** Violin plots of M2 genes expression in the different AM subpopulations.

IR5M₁₀Gy (Supplementary Fig. 4d). Moreover, the spatial distribution of AMs showed major changes in IR5M₁₇Gy. While in the NI control and IR5M₁₀Gy lungs individual AMs are localized within the alveolar compartment, enlarged AMs are gathered in clusters in the fibrotic tissue 5 months after IR₁₇Gy (Supplementary Fig. 4d).

### Endothelial cells (ECs) undergo strong transcriptomic changes specifically after fibrogenic doses of IR

Precise annotation of 6,482 ECs from the NI, IR₁₀Gy, and IR₁₇Gy samples led to the identification of 5 main compartments: lymphatic ECs, artery ECs, vein ECs, gCap and aCap (Fig. 5a, b). These compartments were defined by the expression of markers already described in the literature (Supplementary Fig. 5a): *Efnb2* and *Fbln5* for artery ECs; *Nr2f2* and *Vwf* for vein ECs; *Mmrn1*, *Fxyd6*, and *Fgl2* for lymphatic ECs, gCaps were defined by the expression of *Ptprb* and *Gpihbp1*, and aCap by the expression of *Fibin*, *Car4*, *Apln*, *Tmcc2*, and *Prx*. Examination of the evolution in the proportions of the different endothelial populations after IR revealed a progressive decrease in the proportion of gCap over

time after IR₁₇Gy (from 55,8% in the NI to 17,3% at IR5M), together with an increase in the proportion of aCap (from 17.9% to 41.6%) (Fig. 5c). Interestingly, this event is not as pronounced in mice irradiated at IR₁₀Gy.

Changes in cell proportions were validated by smFISH experiments that combined *Pecam1*, a canonical marker for endothelial cells, with either *Apln*, for the identification of aCap, or with *Ptprb*, which is expressed in all ECs except aCap and lymphatic ECs (Supplementary Fig. 5a, b). On the one hand, co-staining of NI and IR5M samples did not show any significant change in the proportion of *Pecam1*+ *Ptprb*+ cells 5 months after 10 Gy and 17 Gy IR (Fig. 5d, e). On the other hand, we could observe an increase in the proportion of aCap (*Pecam1* + *Apln* + ) 5 months after IR, both at 10 and 17 Gy (Fig. 5d, e). Moreover, immunohistochemistry of NI and IR5M₁₇Gy lung tissue samples with an Apln antibody confirmed this increase in aCap cells after radiation injury (Supplementary Fig. 5c). Therefore, these results confirm the increase in the aCap population observed by scRNA-seq.

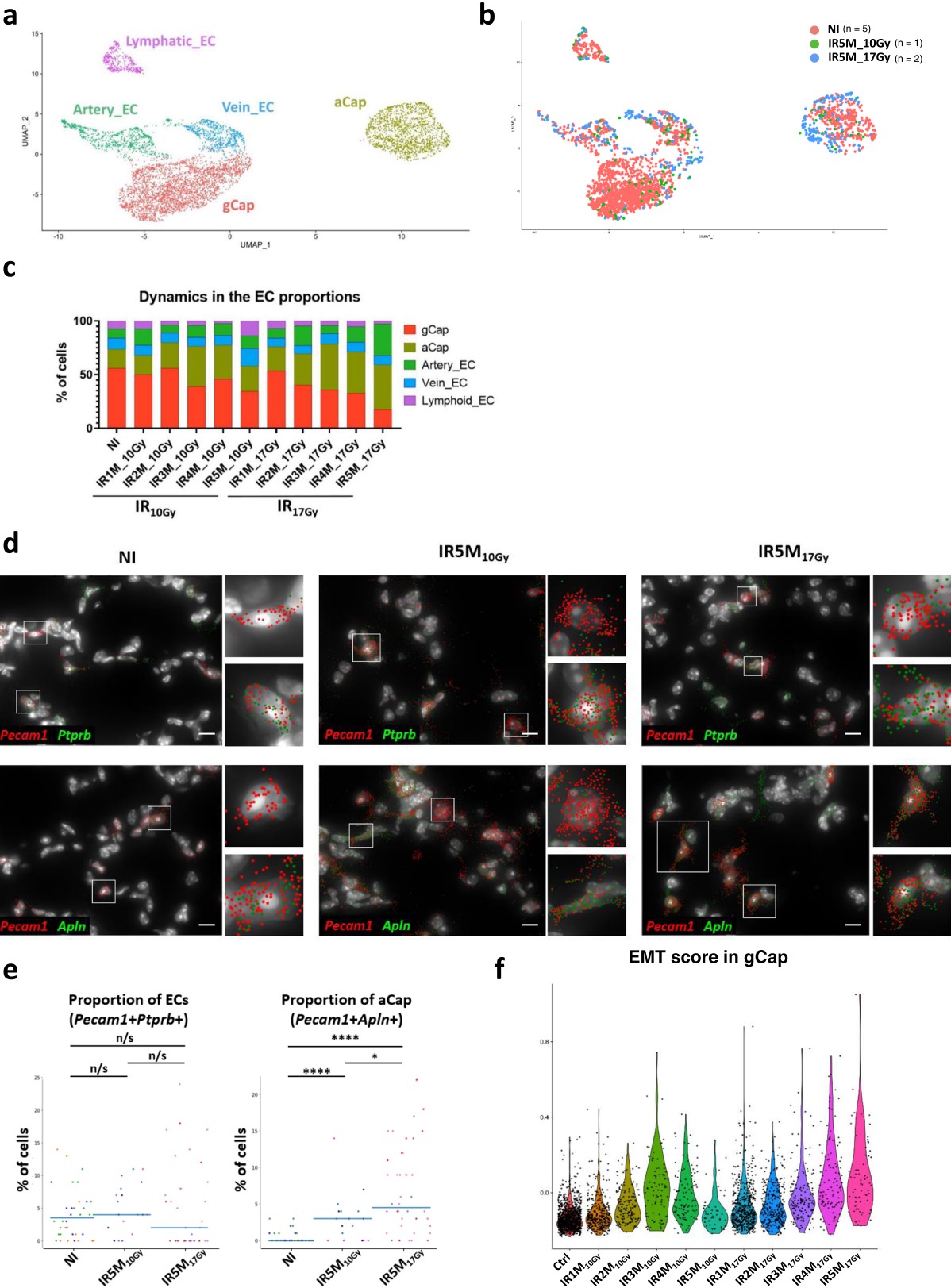

DEG analysis of irradiated capillary ECs versus control revealed a very strong gene deregulation induced at 5 M after 17 Gy while the 5 M response induced by 10 Gy was quite limited (Supplementary Fig. 5d). When we looked for signatures, the 17 Gy irradiation response in gCap was associated with a large deregulation of genes involved in endothelial to mesenchymal transition (EndoMT) (e.g.,

*Col4a2, Col4a1*). We generated an EndoMT single-cell score based on the 200 genes of the EMT signature in the GSEA and observed a progressive increase of the score in the months following IR$_{17Gy}$, reaching its maximum 5 M post-IR (Fig. 5f). On the contrary, after IR$_{10Gy}$ there was an initial increase up to 3 M post-IR and a consecutive decrease back to the NI levels (Fig. 5f). The EndoMT

**Fig. 5 | Characterization of the ECs after radiation injury. a** UMAP visualization of 6482 cells from the different EC subpopulations annotated by cell type. **b** UMAP visualization of NI ($n = 5$), IR5M$_{10Gy}$ ($n = 1$) and IR5M$_{17Gy}$ ($n = 2$) ECs annotated by time point. **c** Dynamics in the proportion of the EC subpopulations at the different time points after IR$_{10Gy}$ and IR$_{17Gy}$. **d** Automatic *Pecam1* (red), *Ptprb* (green), and *Apln* (green) mRNA detection with Big-FISH in NI, IR5M$_{10Gy}$, and IR5M$_{17Gy}$ lung tissue sections. Scale bars, 10 μm. **e** Quantification of the *Pecam1 + Ptprb +* and *Pecam1 + Apln +* cells in the NI, IR5M$_{10Gy}$ and IR5M$_{17Gy}$ lung tissue sections. To

compare two groups, the *P* value was computed with the Mann–Whitney–Wilcoxon test (two-sided test) from scipy (n/s, adjusted *p* value >0.05; *, adjusted *p* value <0.05; **, adjusted *p* value <0.01; ***, adjusted *p* value <0.001; ****, adjusted *p* value <0.0001). Each dot represents one analyzed image. Each color per time point represents a different biological replicate (NI $n = 3$; IR5M$_{10Gy}$ $n = 3$; IR5M$_{17Gy}$ $n = 3$). **f** Violin plot showing the single cell score calculated based on the EMT signature from the GSEA in the gCap at the different time points after IR$_{10Gy}$ and IR$_{17Gy}$.

---

signature appears to be more robust in the gCap compartment than in the aCap (Supplementary Fig. 5e, f).

## Cell-cell interaction analysis underlines the importance of the collagen pathway in the evolution of RIPF

To predict which cellular alterations could be more directly implicated in the development of RIPF, we investigated the evolution of potential cell-cell communications in response to irradiation. We used CellChat[24] to pinpoint, based on the scRNA-seq data and the expression of ligands and receptors, cell types that could be interacting at a specific moment. First, we determined the number of potential existing interactions in the different conditions. This approach indicated an increase in cell-cell communications after IR, this increase being higher after 17 Gy than 10 Gy (Supplementary Fig. 6a). The analysis of the state of communication between major cellular compartments showed that the interactions between mesenchymal cells and endothelial cells increased over time, in particular at the latest time points (3 M, 4 M, and 5 M) after IR$_{17Gy}$ compared to NI (Supplementary Fig. 6b) and to IR$_{10Gy}$ (Fig. 6a).

Further refinement of this analysis suggested an increased communication between, on one side, fibroblasts *Col14a1* and myofibroblasts (acting as sources), and, on the other side, gCap cells (being the target) (Fig. 6b). Supplementary Fig. 6c illustrates the relative force of the registered changes affecting interactions between different cell compartments. Pathway communication analysis from fibroblasts *Col14a1* and myofibroblasts to gCap identified multiple pathways that were increased 5 months after IR$_{17Gy}$ compared to IR$_{10Gy}$ (Fig. 6c) (e.g., *Collagen, Fn1, Angptl, Vegf*). We further focused on the study of the collagen pathway as it shows a progressive increase in strength after 3 M in both mesenchymal cell types (Supplementary Fig. 6d). Next, we identified specific pairs of ligands and receptors that were increased after IR$_{17Gy}$ and absent in NI and IR$_{10Gy}$ at the latest time points after IR. This analysis revealed the ligands *Col1a1* and *Col1a2* to be upregulated in fibroblasts *Col14a1* and myofibroblasts and the receptor *Itga3* to be upregulated in the gCap (Fig. 6d, Supplementary Fig. 6e). These analyses support the role of the collagen pathway, more concretely through *Col1a1-Itga3* and *Col1a2-Itga3* interactions connecting mesenchymal cells (matrix fibroblasts and myofibroblasts) to capillary endothelial cells (gCap) during the development of RIPF (Fig. 6e).

## An interactive web-based interface to study lung responses to irradiation

This work describes some of the key features occurring after radiation injury in the lung at the cellular and molecular level. Nevertheless, due to a large amount of generated data, not all the cell types were described, nor all the molecular alterations that occur during fibrogenesis were studied. For this reason, we have built a dedicated website that is accessible to the scientific community, so that anyone can explore our murine single-cell atlas of the lung response to radiation injury and use it for their own research (Supplementary Fig. 7). This open-access website (https://lustra.shinyapps.io/Murine_RIPF_Atlas/), built using the R package ShinyCell[25], allows the investigation of all the different lung cell populations, as well as the changes in gene expression after the different time points and doses of IR.

The website offers a UMAP visualization of the data in which the different metadata parameters can be represented, e.g., the main cell compartments (epithelial, myeloid, mesenchymal, lymphoid, and endothelial cells), the 21 described cell types (Fig. 7a) as well as 31 precise sub-cell types, the different conditions (Ctrl, IR1M$_{10Gy}$, IR2M$_{10Gy}$, IR3M$_{10Gy}$, IR4M$_{10Gy}$, IR5M$_{10Gy}$, IR1M$_{17Gy}$, IR2M$_{17Gy}$, IR3M$_{17Gy}$, IR4M$_{17Gy}$, IR5M$_{17Gy}$), the different time points (Ctrl, 1M, 2M, 3M, 4M, 5M) and the dose of IR (NI, 10 Gy, 17 Gy). Moreover, this first *Metadata vs GeneExpr* tab allows the users to visualize both cell metadata and gene expression side-by-side on low-dimensional representations (Fig. 7b), which allows a direct search of the expression of different genes in the different cell compartments. Moreover, the visualization of two cell metadata or two gene expressions side by side on low dimensional representations is also possible with the *Metadata vs Metadata* and *GeneExpr vs GeneExpr* tabs respectively.

This site also allows a straightforward exploration of the differences in cell proportions using the *Proportion plot* tab, in which the proportions of the main cell compartments, the different cell types (Fig. 7c), and sub cell types along the different conditions (i.e., different time points and doses of IR) can be examined. Another interesting feature that this tool offers is the visualization of the co-expression of two different genes in a single UMAP representation–*Gene coexpression* tab–(Fig. 7d), providing also the number and percentage of cells that express both, none, or only one of the genes. In addition, four different visualization methods are available to study the gene expression: Violin plot (Fig. 7e), boxplot, bubble plot (Fig. 7f), and heatmap (*Violinplot/Boxplot* tab and *Bubbleplot/Heatmap* tab), which allow the comparison of the expression of genes along the different conditions, thus, providing insightful information about radiation-induced fibrosis. These multiple options allow a wide range of methods for data visualization that can be adapted to the user's needs.

In conclusion, this web-based interface presents a rich dataset that permits the scientific community to investigate the response of the lung to radiation injury for their features of interest with no need of previous bioinformatic knowledge. Raw data is also available at GSE211713 for deeper analysis.

## Discussion

We have built a murine single-cell atlas illustrating the early and late responses of the lung to radiation. The dataset contains more than 100,000 cells from 20 different lungs from non-irradiated as well as IR$_{10Gy}$ and IR$_{17Gy}$ mice. Cell type annotations based on published studies identify the 21 main lung cell populations already described using scRNA-seq approaches[26–28]. This study revealed, in each lung population, the progressive transcriptional changes occurring in the months following radiation injury from the acute inflammatory phase to the development of pulmonary fibrosis after exposure to fibrogenic dose of radiation. The comparison of the molecular alterations induced by a non-fibrogenic (IR$_{10Gy}$) versus a fibrogenic dose (IR$_{17Gy}$) allowed to uncover the physiopathological features of radiation injury in the lung.

This time-course single-cell RNA-seq analysis reproduced the classical features of pulmonary fibrosis such as the loss of AT2 cells, the expansion of myofibroblasts as well as the accumulation of foamy macrophages fostering a pro-inflammatory environment[29–32].

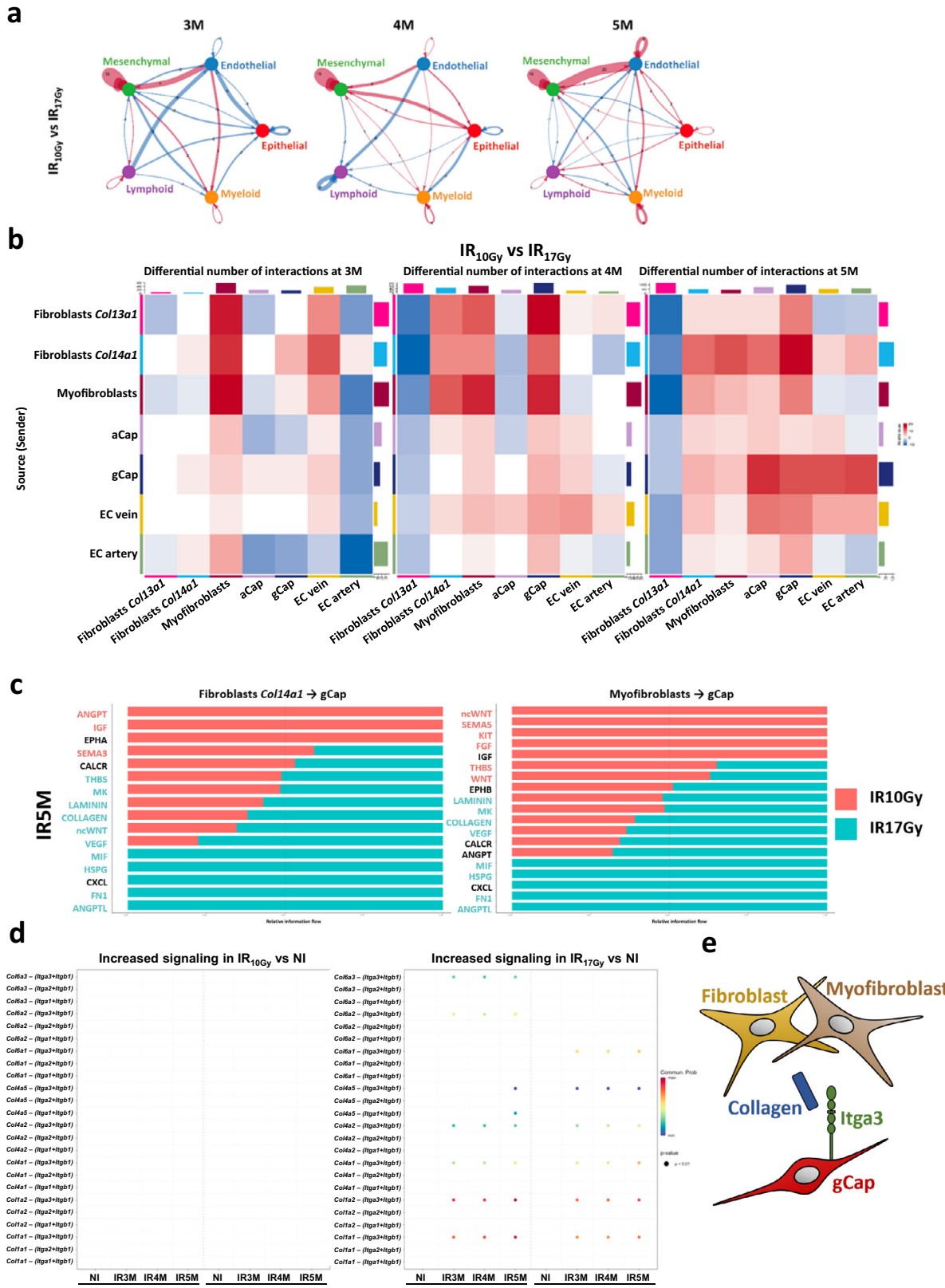

In particular, the pro-fibrotic AM subcluster that we identified under fibrogenic conditions (AM_C2 in our data) may share similarities with a transitional profibrotic macrophage subgroup found in diseased mice using the bleomycin-induced-lung injury model[33]. These transitional macrophages showed an expression profile intermediate between monocyte-derived macrophages and AM that are localized in the fibrotic niche. Moreover, the pro-fibrotic AM expresses genes that have been related to a "foam" phenotype, which corresponds to lipid-laden polarized M2 macrophages that accumulate oxidized phospho-lipids. These foamy macrophages have been found in different mouse models of lung-induced injury as well as in patients with fibrotic lung diseases[30]. Interestingly, our spatial smFISH analysis in lung tissue after

**Fig. 6 | Cell-cell interaction analysis between the different lung cell populations after different time points and doses of IR. a** Circle plot showing the differential number of interactions between $IR_{10Gy}$ and $IR_{17Gy}$ in the main cellular compartments at 3 M, 4 M, and 5 M post-IR: mesenchymal, endothelial, epithelia, myeloid and lymphoid. Red (or blue) colored edges represent increased (or decreased) signaling in the $IR_{17Gy}$ compared to the $IR_{10Gy}$. **b** Heatmap showing the differential number of interactions between $IR_{10Gy}$ and $IR_{17Gy}$ in the endothelial and mesenchymal subpopulations at 3 M, 4 M, and 5 M post-IR. Red (or blue) represents increased (or decreased) signaling in the $IR_{17Gy}$ compared to the $IR_{10Gy}$. The top-colored bar plot represents the sum of column of values displayed in the heatmap (incoming signaling). The right-colored bar plot represents the sum of row of values (outgoing signaling). **c** Bar graph showing significant signaling pathways ranked based on differences in the overall information flow within the inferred networks between $IR5M_{10Gy}$ and $IR5M_{17Gy}$ from the Fibroblasts *Col14a1* and Myofibroblasts (sources) to the gCap (targets). The top signaling pathways colored red are enriched after $IR5M_{10Gy}$, and the ones colored green were enriched after $IR5M_{17Gy}$. **d** Increased signaling ligand-receptor pairs of the Collagen pathway in $IR_{10Gy}$ and $IR_{17Gy}$ compared to NI at 3 M, 4 M, and 5 M after IR. **e** Schematic drawing of the intercellular communication between fibroblasts and myofibroblasts with the gCap through the collagen pathway.

radiation injury indicates that these foamy macrophages tend to aggregate in fibrotic foci with an increased cell volume and a high level of endogenous fluorescence. These observations corroborate with data from other groups showing that clusters of M2 macrophages are detected after IR, contribute to fibrosis development, and are controlled by adenosine signaling[34,35].

Concomitantly to the bronchiolization and the destruction of alveoli observed during fibrosis development, the progressive disappearance of AT2 is associated with profound changes in the transcriptome of remaining AT2 cells. Apart from an increased expression of genes associated with EMT, we identified, among the AT2 population after IR, an upregulation of AT1 marker genes, suggesting that AT2 cells are differentiating into AT1 cells. Detailed analysis confirmed an enrichment in Krt8+ transitional AT2 cells, known to support alveolar regeneration[17]. The fact that AT2 cells are known to give rise to AT1 cells[36] and that, after lipopolysaccharide (LPS)-induced lung injury, AT2 cells activate a similar transdifferentiation program[16] indicate that a common regenerative process is activated after IR to restore normal alveolar architecture and lung function. Considering that the transdifferentiation program is turned off in the months following exposure to a nonfibrogenic dose of radiation whereas, after a fibrogenic dose, AT2 cells maintain and reinforce this transdifferentiation program, it is tempting to speculate that failure to differentiate into functional AT1 cells leads to AT2 cells exhaustion, contributing to alveoli destruction and fibrosis development. However, more in-depth analysis of AT2 after radiation injury is required to demonstrate a direct physiopathological implication in fibrosis development after IR.

Endothelial cells are particularly affected by radiation[37]. Our single-cell RNA-seq analysis showed that endothelial cells activate an EndoMT program in the months following exposure to radiation. It is known that EndoMT participates in many human fibrotic disorders[38] and a previous study identified hypoxia and TGFβ as EndoMT inducers after radiation injury[39]. It has been suggested by Hashimoto and colleagues that endothelial cells can give rise to myofibroblasts in the mouse model of pulmonary fibrosis induced by bleomycin[40]. Interestingly, cell-cell communication analysis after radiation injury pointed to increased signaling between endothelial cells and stromal cells, particularly myofibroblasts. These results support the idea that radiation triggers EndoMT phenotype coupled to increased interactions with myofibroblasts, suggesting that endothelial cells may contribute as a source in the expansion of myofibroblasts during the development of RIPF.

In summary, this study describes the transcriptional changes occurring at the single-cell level in the lung in the months following radiation injury. This large and comprehensive dataset provides a starting resource to explore and characterize molecular mechanisms underlying the different physiological and pathological steps occurring after IR (e.g., from acute wound healing to fibrosis development). To facilitate data visualization and interrogation, a user-friendly web-based interface is accessible to the scientific community without the need for specific computational skills.

## Methods

### Mice and ethics statement
Studies were performed in accordance with the recommendations of the European Community (2010/63/UE) for the care and use of laboratory animals. Experimental procedures were specifically approved by the ethics committee of the Institut Curie CEEA-IC #118 (Authorization number APAFIS#5479-201605271 0291841 given by the National Authority) in compliance with the international guidelines. Females C57BL/6 J mice purchased from Charles River Laboratories at the age of 6 weeks were housed in Institut Curie animal facilities.

### Radiation injury
Here, the classical C57BL/6 J female mouse model of lung radiation toxicities were used[11]. Collimation, time-resolved fluence measurement, chemical dosimetry, depth-dose distribution, anesthesia of the mouse, mouse immobilization, and irradiation of mouse thorax were carried out as previously described[41,42]. Briefly, mice were anesthetized with a nose cone using 2.5% isoflurane in air, without adjunction of oxygen, immobilized in a dorsal position, and set in a vertical position at 500 mm of the electron source. Then, mice were exposed to bilateral thorax irradiation with a dose of 10 or 17 Gy at the age of 10–12 weeks using the 4.5-MeV linear electron accelerator facility (Kinetron).

In order to determine the level of pulmonary fibrosis in mice, the lung was imaged with three-dimensional X-ray on the cone beam computed tomography (CBCT) module of the Small Animal Radiation Research Platform (SARRP, Xstrahl). 1.5–2% isoflurane was used to anesthetize the mice, which were maintained on a PMMA vertical stand in the vertical upright position. The 3D reconstruction of the images was calculated from 1,440 projections using the integrated software Murislice (XStrahl). Then, ImageJ/FIJI (ImageJ, NIH, Bethesda, MD) was used to reconstruct the slices. This analysis allowed to determine the level of fibrosis of the mice. Further details can be found in ref. 42.

### Lung tissue dissociation
Mice were killed by cervical dislocation and the ribcage was opened to clear the trachea. Mouse trachea was perfused with 1.5 ml of 50 U/ml dispase (Serlabo, WO-LS02100; Sigma Corning, DLW354235) using a 20 G needle, followed by 0.5 ml of 1% agarose (Invitrogen, 15510-027) to block the exit of the dispase. Lungs were resected, minced with a scalpel into small pieces, and added into 3 ml of 1× DPBS $MgCl^{2+}$ and $CaCl^{2+}$ (Gibco, 14040-091). Then 320 µl of 25 U/ml elastase (Worthington, LS002292) were added and the suspension was homogenized and incubated for 30 min at 37 °C with orbital shaking. Enzymatic activity was inhibited with 5 ml of PF10 (1× DPBS containing 10% fetal bovine serum (FBS)) and 20 µl of 0,5 M EDTA pH 8 (Invitrogen, AM9260G). Cell suspension was filtered through 100 µm nylon cell strainer (Fisher Scientific, 22363549), which was rinsed with 5 ml of PF10. This was followed by 37.5 µl of 10 mg/ml DNase I (Sigma, D4527-40KU) treatment and incubation on ice for 3 min. Cell suspension was filtered again through a 40 µm nylon cell strainer (Fisher Scientific, 087711) and 5 ml of PF10 were added to rinse it. Samples were centrifuged for 6 min at 150 g and 4 °C, pellet was resuspended in red blood cell (RBC) lysis buffer (Roche,

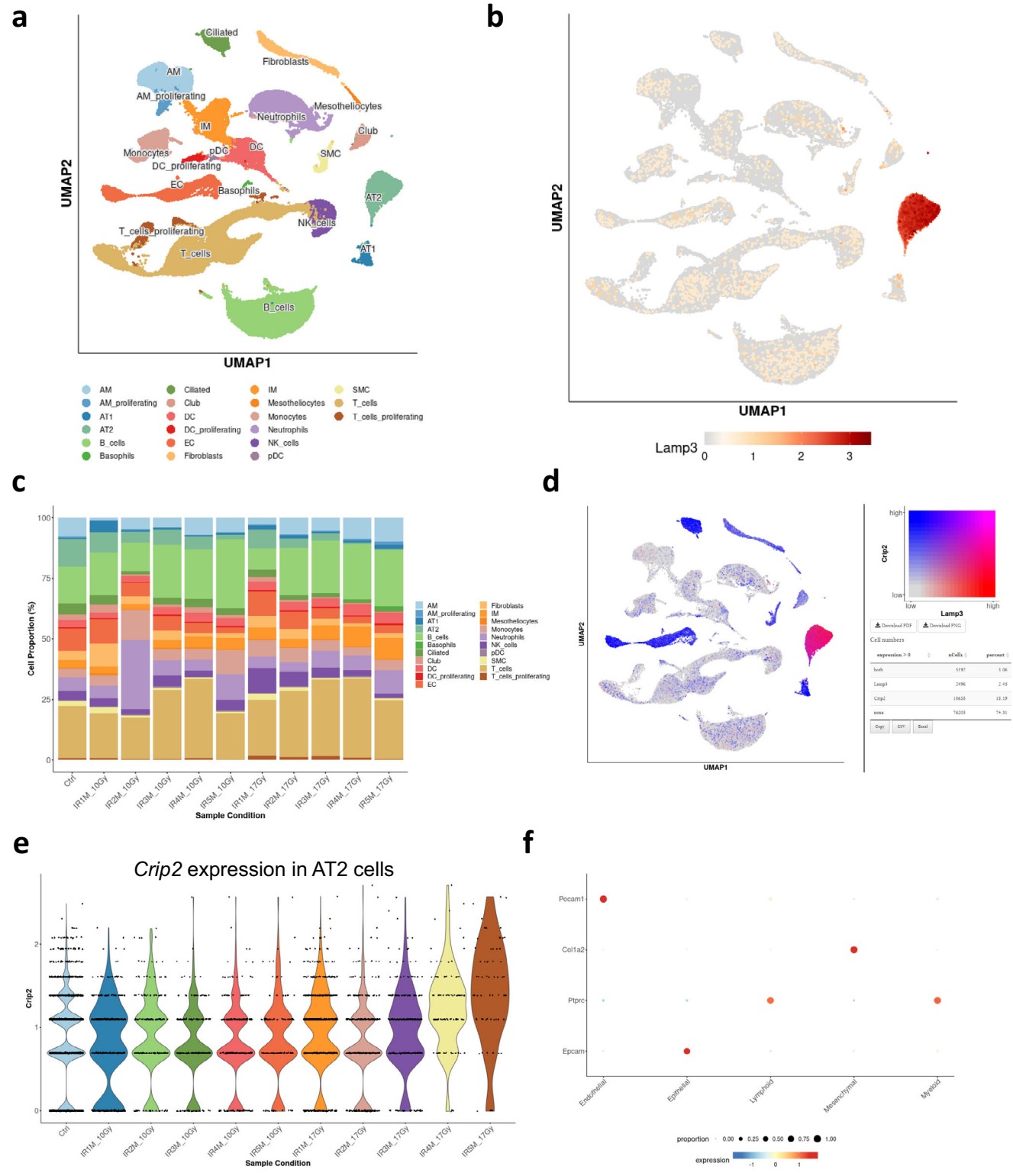

**Fig. 7 | Web-based interface for the murine single-cell atlas of the lung response to radiation injury. a** UMAP visualization of 102,869 cells from 20 different samples (5 NI; 5 IR$_{10Gy}$, one per time point; 10 IR$_{17Gy}$, two per time point) annotated by cell type. **b** UMAP visualization of the expression of *Lamp3*. **c** Dynamics in cell proportions of the main cell types across the NI and IR conditions at the different time points and doses. **d** UMAP visualization of the expression of *Lamp3* (red), *Crip2* (blue), and the co-expression of both (pink). **e** Violin plot of *Crip2* expression in the AT2 cells in the NI samples and at the different time points after IR$_{10Gy}$ and IR$_{17Gy}$. **f** DotPlot of the expression of the marker genes used to identify the main 5 cell compartments.

11814389001) and incubated for 90 s at room temperature (RT) before stopping the lysis with 6 ml of PF10. Then, 500 µl of pure FBS were placed at the bottom of the sample, prior to a final centrifugation for 6 min at 150 × *g* and 4 °C. The pellet was resuspended in 1 ml of 1× DPBS containing 0.02% bovine serum albumin (BSA) (Sigma, D4527-40KU), and cell counting was done in a Malassez. Finally, concentration of the samples was adjusted to 1 million cells/ml in 1× DPBS containing 0,02% BSA.

## Droplet based scRNA-seq (10x GENOMICS)

Single-cell 3′-RNA-Seq samples were prepared using single cell V3.1 reagent kit and loaded in the Chromium Controller according to standard manufacturer protocol (10x Genomics, PN-120237) to capture 6,000 cells. Briefly, dissociated lung single cells are encapsulated in nanodroplets (GEMs) using a microfluidic device. These GEMs are generated combining barcoded single cell 3′ V3.1 gel beads, a master mix that contains the reverse transcription (RT) reagents, the single cells, and partitioning oil onto the Chromium Next GEM Chip. After cell lysis, RNAs are captured on the gel beads coated with oligos containing an oligo-dTTT, unique molecular identifiers (UMIs) and a specific barcode.

Incubation of the GEMs produces barcoded, full-length cDNA from poly-A mRNA. After reverse transcription, GEMs are broken and cDNAs are purified with silane magnetic beads. Then, barcoded full-length cDNA is amplified by PCR to generate enough material for library construction. Amplified cDNA is purified again, and cDNA quality control was assessed by capillary electrophoresis (Bioanalyzer, Agilent) before the preparation of the libraries.

Finally, libraries are prepared using a fixed proportion of the total cDNA. Enzymatic fragmentation and size selection are used to optimize the cDNA amplicon size. During the GEM incubation, the read 1 primer sequence is added to the molecules. At this step, P5, P7, a sample index, and the read 2 primer sequence are added via End Repair, A-tailing, Adaptor Ligation, and PCR. This way, the final libraries contain the P5 and P7 primers used in Illumina bridge amplification. Finally, libraries were sequenced on a NovaSeq sequencer (Illumina). Each measurement comes from independent NI, IR$_{10Gy}$, or IR$_{17Gy}$ mice.

## scRNA-seq data analysis

First, raw sequencing was processed using the 3.0.2 Cell ranger pipeline (10x Genomics). Briefly, Illumina sequencing files (bcl2) were demultiplexed and mapped onto the mm10 reference genome. This allows the creation of a count matrix table for each of the samples. Then, the count matrices were individually loaded in R (4.0.5) and analyzed using Seurat package v4.0.1[43].

In order to remove the contamination from the "soup" of cell-free RNAs, we used SoupX[44]. This tool is able to remove ambient RNA contamination from droplet-based scRNA-seq experiments. Briefly, first SoupX estimates the mRNA expression profile from empty droplets. Then it estimates the contamination fraction and the fraction of UMIs originating from the background in each cell. Finally, it corrects the expression of each cell using the ambient mRNA expression profile and estimated contamination.

SoupX matrices were imported in Seurat, and a Seurat object was created for each of the samples with the function CreateSeuratObject. Samples were merged and the object was normalized using the SCTransform normalization function, a normalization and variance stabilization method using regularized negative binomial regression[45]. Then, PCA was performed and the first 20 PCs were selected (based on inspection of PC elbow plot) as input for RunUMAP function for dimension reduction and visualization with the Uniform Manifold Approximation and Projection (UMAP) dimensional reduction technique[46]. Finally, cells were clustered with the functions FindNeighbors and FindClusters, based on the default Seurat parameters with resolution parameter set to 0.8. In a first step, we performed general QC: gene counts matrices were filtered and cells with nCounts <200 and nFeatures >6000 RNA molecules sequenced, as well as percentage of mitochondrial genes >15% were removed. In a second step, each cell population was cleaned in more details by removing the cells that presented a high percentage of mitochondrial genes, a low nCounts, or that expressed markers characteristic of other cell types, which suggests the presence of doublets. All quality controls are present on the web interface.

Then, cell-type marker genes for each cluster were identified using the function FindAllMarkers of Seurat. Cell clusters were named based on cell type-specific markers from recently published scRNA-seq datasets[12–14]. For each of the analyzed clusters, we performed differential expression analysis using the function FindMarkers with the MAST package (two-sided test), which identifies differentially expressed genes between two groups of cells using a hurdle model tailored to scRNA-seq data[47]. We studied the genes with a logFC threshold of >1 or <−1 and an adjusted p-value threshold of <0.05 using the Gene Set Enrichment Analysis (GSEA) computational method, which defines if a set of genes shows statistically significant, concordant differences between two states.

Violin plots and Heatmaps were generated with Seurat for specific significant differentially expressed genes. To calculate the single cell data score, first a dataset corresponding to a cell state or biological process is chosen. Then, the table of expression of these genes by each cell is extracted. For each gene, the expression data is centered and normalized. Finally, for each cell, we computed the mean of the centered normalized gene expression. This gave us an expression score for each cell which was then plotted with UMAP. The trajectory analysis of the AT1 and AT2 cells was performed using Monocle 3[48]. The pseudotime analysis of the AT1 and AT2 cells was done with Monocle 2[49], where the marker genes of the AT2 cluster 3 were used to order the cells. The gene regulatory network analysis was constructed on the AT2 using SCENIC[50]. Lastly, all cell-cell interactions analysis were done using CellChat[24].

## Mouse lung tissue processing for smFISH

Mice were injected with a mix of 100 mg/kg ketamine and 10 mg/kg xylazine and we waited for the mice to be fully asleep. Then, the ribcage was opened, the heart was perfused with 10 ml of cold 1× PBS pH 7.4 (Invitrogen, AM9624) and it was resected just after the perfusion was finished. Then, the mouse trachea was perfused with cold 4% paraformaldehyde (PFA) (Euromedex, 15714-S) until the lungs were fully expanded with no air left inside. The trachea was closed with a thread to avoid PFA leakage. The lungs were resected out of the ribcage and kept in a falcon with cold 4% PFA overnight (o/n) under rotation at 4 °C. After fixation, the 5 lobes were separated and kept individually in cold 1× PBS containing 30% sucrose (Sigma, S7903) during 6 h under rotation at 4 °C. Lobes were rinsed in cold 1× PBS and pre-embedded in cold 50% optimal cutting temperature (OCT) compound (VWR, 411243) diluted in 1× PBS during 30 min under rotation at 4 °C. Finally, each lobe was embedded in square embedding molds (VWR, POLS18646ACODE45) containing OCT, frozen in dry ice during 20 min and stored at −80 °C.

## smFISH probes design and preparation

The design of smFISH probes was performed with the R-package Oligostan[51]. For a given target mRNA, Oligostan outputs a list containing all the potential probe sequences fulfilling these requirements: length between 26 nt and 32 nt, score around $\Delta G_{37\,°C}$ value of 90%, minimal distance between probes of 2 nt, GC percentage between 40 and 60%, 5 different criteria for probe composition (nucleotide composition in A < 28%, no AAAA stacks, C nucleotide composition between 22 and 28 %, no CCCC stacks in any six consecutive nucleotides in the first 12 positions and no 4 nonconsecutive C in any 6 consecutive nucleotides in the first 12 positions). To each RNA-specific sequence, a shared readout sequence (FLAP Y, 27 nt) was added, and probes with the highest scores were selected. To obtain a good smFISH signal, around 30 probes per RNA were used (Lamp3 (31), Chil3 (32), C3ar1 (32), Pdgfra (32), Hhip (32), Pecam1 (32), Ptprb (32), Apln (32), Fibin (28), Prx (32), Tmcc2 (29))[51].

For visualization, we used secondary probes carrying two fluorophores (either Cy3 or Cy5) on both ends through 5′ and 3′ amino modifications: FLAP Y-Cy3 (/5Cy3/AA TGC ATG TCG ACG AGG TCC

GAG TGT AA/3Cy3Sp/) and FLAP Y-Cy5 (/5Cy5/AA TGC ATG TCG ACG AGG TCC GAG TGT AA/3Cy5Sp/).

smFISH primary probes and secondary probes (fluorescent FLAP probes) were produced and bought from Integrated DNA Technologies (IDT). Primary probes are delivered frozen in 96-well plates at a final concentration of 100 μM in Tris-EDTA pH 8.0 (TE) buffer. An equimolar mixture of the primary probes was prepared per set of probes and diluted to 20 μM in TE buffer pH 8 (Invitrogen, AM9849). Secondary fluorescent FLAP probes are delivered lyophilized. They were resuspended in TE buffer at a final concentration of 100 μM. Primary and secondary probe stocks were stored at −20 °C.

### smFISH on lung tissue sections

OCT-embedded mouse lung lobes were cut into 16 μm tissue sections in a cryostat (Leica CM 1950) and mounted in previously cleaned and coated coverslips. Briefly, extensive cleaning was achieved by washing the coverslips (Menzel-Glaser, 20 × 20, #1) three times in ethanol (VWR, 20821.310), acetone (Honeywell, 32201), and water. Then, they were sonicated in 1 M KOH (Honeywell, 06005) in $H_2O$, rinsed with water and dried in an oven at 70 °C for 10 min. For tissue attachment, coverslips were then coated with 2% (3-aminopropyl)triethoxysilane (APTS) (Sigma, A3648-100ML) in $H_2O$ for 2 min and rinsed with water twice. Finally, the coating was activated in the oven at 70 °C for 60 min. Coverslips with lung tissue sections were placed into 6 well plates, fixed with cold 4% PFA for 15 min at 4 °C, washed twice with cold 1× PBS, and kept in 70% ethanol at 4 °C o/n.

smFISH probes were generated by pre-hybridizing the primary probes with the secondary FLAP Y (FLAP Y-Cy3 or FLAP Y-Cy5) probes in a PCR machine. Briefly, a mix containing 40 pmol of the primary probes, 50 pmol of the FLAP Y probe, 1× NEBuffer 3 (BioLabs, B7003S) and ultra-pure water was incubated in a thermocycler for 3 min at 85 °C, 3 min at 65 °C and 5 min at 25 °C. The hybridization mix was prepared by mixing 2 μl of the smFISH probes and 98 μl of hybridization buffer (100 mg/ml dextran sulfate (Sigma, D8906-10G) and 10% deionized formamide (Invitrogen, AM9342) in 2× SSC (Invitrogen, AM9763)) per sample. Hybridization buffer can be prepared in advance and frozen in aliquots. These aliquots were thaw and heated to 100 °C for 5 min, and let cool to RT. In case of co-staining, 96 μl of hybridization buffer was mixed with 2 μl of primary probes hybridized to FLAP Y-Cy3 and 2 μl of primary probes hybridized to FLAP Y-Cy5.

For the hybridization of the smFISH probes in the tissue, first tissue sections were re-hydrated twice with washing buffer I (WBI) (2× SSC) for 3 min at RT, followed by a last incubation in washing buffer II (WBII) (10% deionized formamide in 2× SSC) for 3 min at RT. The hybridization of the smFISH probes in the lung tissue sections was carried out in a hybridization chamber. Here, the tissue sections were placed upside-down onto a 100 μl droplet of the hybridization mix and incubated at 37 °C o/n.

To finish, tissue sections were washed three times: first in preheated WBII at 37 °C for 30 min, second in WBII containing 0.05 μg/μl DAPI (Sigma, D9542-1MG) at 37 °C for 30 min, and third in 1× PBS for 5 min at RT. Samples were mounted in SlowFade Diamonds (Invitrogen #S36963) and stored at 4 °C until image acquisition.

### Image acquisition

For each sample, 54 three-dimensional image stacks were captured with an interval of 0.3 μm (total of 16.2 μm) on a widefield microscope (Upright Widefield Apotome Zeiss) equipped with a 63 × 1.4 numerical aperture (NA) objective and a CCD camera (CoolSNAP HQ2) and controlled with ZEN microscope software (ZEISS Microscopy). Three lasers were used to excite DAPI (excitation time 40 ms) and smFISH probes labeled with Cy3 (excitation time 300 ms) and Cy5 (excitation time 300 ms). When acquiring tiles with a 5 × 5 FOV, 16 three-dimensional image stacks were captured with an interval of 0.3 μm (total of 4.8 μm).

### Image analysis

Image analysis and result generation are performed automatically with a provided command line script performing the steps described next.

Nuclei were automatically segmented with Cellpose[52] and RNA detection was performed with FISH quant v.2[53]. Points detected in several channels at the same location were considered to be the result of auto-fluorescence and therefore removed.

Custom written Python scripts were used to determine the cell-type of a nucleus. Importantly, marker genes were chosen such that each cell type is identified with only one marker gene. First, RNAs of each gene were clustered with the algorithms OPTICS and DBSCAN[54] in their implementation in scikit-learn (https://scikit-learn.org/). We used the following functions and parameters:

`sklearn.cluster.OPTICS` is used to determine the core points and ordering. Parameters are `min_sample=4` and `min_cluster_size=4`.
`sklearn.cluster.cluster_optics_dbscan` is used for the clustering, with the parameters obtained in the previous step. The parameter eps was manually adjusted for each gene to account for differences in the expression levels. Other parameters are left to their default value.

The next step consisted in assigning the detected RNA clusters to the individual cells (or nuclei, as there is no cytoplasmic marker). For this, we calculated the convex hull for each individual cell with Delaunay tessellation (using spatial.Delaunay from scipy). In order to assign a convex hull to a nucleus, we required that there was at least an overlap of $K$%, where $K$ was set individually for different cell types (20 to 55%) to account for different cell morphologies. Non-assigned convex hulls were removed from the analysis (this typically happened when nuclei were not present in the image). The proportion of positive cells for a specific cell type marker in an image was calculated as the number of positive nuclei of this marker over the total number of nuclei segmented in the image. To compare two groups (ex NI vs IR5M), the $P$ value was computed with the Mann–Whitney–Wilcoxon test (two-sided test) from scipy (*$P < 0.05$; **$P < 0.01$; ***$P < 0.001$, ****$P < 0.0001$). Cell volume was estimated as the volume of the convex hull associated to a nucleus (Fig. 2b). If a point cloud contained several nuclei (for instance for neighboring cells of the same type), the average cell volume per nuclei was calculated. For the violin plots, each independent NI, $IR_{10Gy}$, or $IR_{17Gy}$ mice are represented with different colors, and dots of the same color represent different analyzed images from the same mouse sample.

### Reporting summary

Further information on research design is available in the Nature Portfolio Reporting Summary linked to this article.

## Data availability

The scRNA-seq datasets generated during the current study have been deposited in the Gene Expression Omnibus (GEO) repository, with the accession code GSE211713. Processed data can be explored through an interactive web interface (https://lustra.shinyapps.io/Murine_RIPF_Atlas/). Source data are provided with this paper.

## Code availability

We make the full image analysis Python code available on the following Github page https://github.com/tdefa/cell_type_calling_2channels and on Zenodo https://zenodo.org/record/7360791 (https://doi.org/10.5281/zenodo.7360791). Other codes are available from the authors upon kind request.

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

## Acknowledgements

This work has received financial support through an Agence Nationale de la Recherche (ANR) grant (Lustra to A.L.-V., F.Mu., T.W., and C.F.). It has also received support from EDF (to A.L.-V. and C.F.), La Ligue Contre Le cancer (to A.L.-V. and S.C.A.), the Institut National du Cancer (INCa) (to A.L.-V.), the Institut Curie ICGEx granting program (to A.L.-V.) and the European Union's Horizon 2020 research and innovation program under the Marie Skłodowska-Curie grant agreement No 666003 (to S.C.A.). Furthermore, this work was supported by the French government under the management of Agence Nationale de la Recherche as part of the "Investissements d'avenir" program, reference ANR-19-P3IA-0001 (PRAIRIE 3IA Institute to T.W.). F.Mu. and C.W. acknowledge funding by Institut Pasteur. The authors greatly acknowledge the Multimodal Imaging Center-Light Microscopy Facility of the Institut Curie (CNRS UMS2016/InermUS43/Institut Curie/Université Paris-Saclay), as well as the Cell and Tissue Imaging Platform—PICT-IBiSA (member of France–Bioimaging—ANR-10-INBS-04) of the U934/UMR3215 of Institut Curie for help with light microscopy. The authors wish to thank Christophe Alberti, Elodie Belloir, Cédric Lantoine, and Virginie Dangles-Marie from the animal core facility of Institut Curie. The contribution of the bioinformatics core facility at U900 Inserm-Institut Curie is gratefully acknowledged. High-throughput sequencing was performed by the ICGEx NGS platform of the Institut Curie supported by the grants ANR-10-EQPX-03 (Equipex) and ANR-10-INBS-09-08 (France Génomique Consortium) from the Agence Nationale de la Recherche ("Investissements d'Avenir" program), by the ITMO-Cancer Aviesan (Plan Cancer III) and by the SiRIC-Curie program (SiRIC Grant INCa-DGOS-465 and INCa-DGOS-Inserm_12554). Data management, quality control, and primary analysis were performed by the Bioinformatics platform of the Institut Curie.

## Author contributions

S.C.A., C.F., C.W., H.L., S.Le, and S.La performed the experimental studies. S.C.A., J.S., C.F., T.D., T.W., and F.Mu. carried out the computational analysis of scRNA-seq data and smFISH images. S.H. built the web interface. S.C.A., J.S., T.D., T.W., F.Mu, J.A.L.V., and C.F. participated in the writing of the paper. J.A.L.V. and C.F. supervised the work. S.C.A., J.S., T.D., C.W., S.H., H.L., S.Le, S.La, M.D., V.F., F.Ma, T.W., F.Mu, J.A.L.V., and C.F. contributed to data discussion.

## Competing interests

The authors declare no competing interests.
