## [Peer Review File · Nature Communications]

An interactive murine single cell atlas of the lung responses to radiation injuryReviewer #1 (Remarks to the Author):

This is an interesting paper that I find to be significant and important. This study demonstrated a spatio-temporal single cell atlas of the lung from RILI murine models. It showed detailed cell types and cell-cell communications/interactions that might involve in the progression of RIPF.

The main concern is that the studies defined the cell types according to previously published scRNA-seq cell markers. Because different publication uses different cell markers, the annotation would be biased. It might be worth trying the single cell RNA-seq annotation tools such as Cello, ScType, and ext. that presumably could provide unbiased and automated cell type annotation, to see if this can result in more unbiased cell type identifications or new cell types that are specific to RILI.

For the epithelial compartments, there are recent publications suggest AT2 transition/persistent progenitors/Krt8+ transitional stem cells in the injured lung. It will be interesting to see if the RILI datasets contain this cell population.

For the mesenchymal cell compartments. It will be interesting to see the changes in pericytes and mesothelium cells during RILI progression. Besides, there are recent publications showing the existence of the Ebf1+ invasive fibroblast, not sure if the authors have detected this population of mesenchymal cells or not.

In the methods, the authors didn't mention the sex of the mouse models. The authors should follow the 'Sex and Gender Equity in Research – SAGER – guidelines' and include sex and gender considerations for studies involving vertebrate animals.

Reviewer #2 (Remarks to the Author):

In this study, Sandra Curras-Alonso et al. performed scRNA-seq of dissociated lungs from non-irradiated mice and mice 1, 2, 3, 4 and 5 months after fibrogenic (17 Gy) and non-fibrogenic (10 Gy) doses of IR, which provided a whole organ single cell atlas spanning the evolution over time towards pulmonary fibrosis, from the early response and the inflammatory phase to the late tissue reaction of fibrotic process. Firstly, the data of single cell transcriptome atlas aim at studying radiation-induced pulmonary fibrosis is indispensable resource. Secondly, some experiments, such as smFISH, in addition to sequencing data, were included for validation. Thirdly, an interactive web-based interface is useful to study lung responses to irradiation. Overall, this is a valuable study, but the following points should be addressed.

Major concern:

1. Quality control is important for single cell atlas. The quality measures of scRNA-seq data were insufficient in this study. More quality control analyses of scRNA-seq are needed.
2. Some interesting molecules in transdifferentiation were identified in response to fibrogenic irradiation, it is suggested to construct the regulatory network further from the scRNA-seq.
3. Why not further analyze transdifferentiation by Monocle? Monocle orders individual cells according to progress through a biological process.
4. It would be better if the website supports the scATAC-seq data, for the data of the single cell transcriptome and paired chromatin accessibility atlas are quite precious.

Minor comments:

1. The website built by authors should include a webinterface for data manipulation.
2. Proper formatting must be used; symbols for genes are in italics, and symbols for proteins are

in non-italicized font. Corrections must be reflected in the figures and tables as well as text.

3. "scRNAseq" should be "scRNA-seq", if it is the abbreviation for single-cell RNA sequencing.
4. Page 6 line 173, the description "1M after IR17Gy 30% of these genes..." , suggest to rewrite as "1m after 17 Gy irradiation, 30% of theses genes...".
5. Page 7 line 221, "the first subset is characterized by the expression of Folr2, Ccl8 and Cd163 (called here IM_C1)". The Supplementary Fig. 4a did not show the expression of the Folr2 and please add it.
6. Page 10 line 353, the accession nos. of data for GEO was missing.

We sincerely appreciated the reviewers' time and effort to read and comment on our manuscript. We have addressed every single comment and suggestion, and a point-by-point response is found below. The reviewers' comments appear in black and our responses are in blue. All changes made in the manuscript and figures are highlighted. We do hope this revised version of our manuscript will meet the requirements for publication in Nature Communications.

REVIEWER COMMENTS

Reviewer #1 (Remarks to the Author):

This is an interesting paper that I find to be significant and important. This study demonstrated a spatio-temporal single cell atlas of the lung from RILI murine models. It showed detailed cell types and cell-cell communications/interactions that might involve in the progression of RIPP.

The main concern is that the studies defined the cell types according to previously published scRNA-seq cell markers. Because different publication uses different cell markers, the annotation would be biased. It might be worth trying the single cell RNA-seq annotation tools such as CellO, ScType, and ext. that presumably could provide unbiased and automated cell type annotation, to see if this can result in more unbiased cell type identifications or new cell types that are specific to RILI.

Response: We thank the reviewer for the suggestion and we agree that an unbiased approach for cell type annotation could be useful to identify previously uncharacterized cell types in RILI. As suggested by the reviewer, we have evaluated several single cell RNAseq annotation such as CellO and ScType. However, CellO has been designed for annotation of human datasets and we could not apply it on our mouse data. ScType, on the other hand, could be used and provided, at least in our hands, an annotation that appeared in some instances to be strikingly inconsistent with regards to cell-specific markers very well described in the literature (see Figure 1 below, for this review only). From our experience, the performance of single cell annotation tools is highly dependent of the reference used, that is, the more complete is the reference, the more performant the tool is expected to be. To our knowledge, there is no comprehensive lung single cell reference with precise and validated cell annotations yet, thus preventing a broad use of automatic cell type annotation algorithms, at least in the lung. If such comprehensive lung cell type atlas becomes available in the future (and we hope that our annotations will contribute to that), we will be ready to use it to re-annotate our RILI dataset and update the interactive web interface accordingly. Finally, our manual approach still allows the identification of new cell types, which, by definition, will not be found in any reference. Indeed, it is possible that some of the unbiased lists of specific markers identified as characteristic of particular clusters identified by the Seurat package do not overlap major cell-specific markers already described in the literature. In that case, supplementary analyses (re-clustering of specific cell compartments and differential gene expression, for instance, with well characterized cell types) can provide hints to define the identity of the cells and whether or not these cells are present at the basal state or only arise as a response to injury.

Figure 1 (for review only) : Evaluation of automatic cell annotation by ScType. In A), clustering performed by the Seurat package using our lung RILI dataset followed by either an automatic annotation generated automatically by ScType in B) or our manual annotation based on markers from the literature in C). As an example of performance, we can examine cluster 1, which is annotated as basal cells by ScType and T cells by our method. As shown in D) cluster 1 does not express markers specific of basal cells (i.e. Krt5, Trp63, Dapl1). Instead, it expresses classical markers of T cells (i.e. Cd3g, Cd3e, Trbc2). Similarly, ScType annotated cluster 3 as alveolar macrophages (AM) whereas it does not express canonical markers of alveolar macrophages but is instead characterized by the expression of neutrophils markers. From this simple analysis, we conclude that using automatic annotation tool such as ScType can lead to erroneous annotation.

For the epithelial compartments, there are recent publications suggest AT2 transition/persistent progenitors/Krt8+ transitional stem cells in the injured lung. It will be interesting to see if the RILI datasets contain this cell population.

Response: We thank the reviewer for this important suggestion. We looked into the expression of the Krt8 gene in the AT2 population in our RILI dataset. Interestingly, Krt8 is predominantly expressed in a subset of AT2 (i.e. sub-cluster 5) that is specifically enriched after exposure to a fibrogenic dose of radiation (17 Gy). In accordance with our previous results showing active transdifferentiation of some AT2 cells after 17 Gy (Figure 2), these results further support that transitional (Krt8+) progenitors are mobilized at late timepoints after severe radiation injury leading to lung fibrosis. These results have been included into figure 2 and the text was modified accordingly, with a reference to results published in the literature.

For the mesenchymal cell compartments. It will be interesting to see the changes in pericytes and mesothelium cells during RILI progression. Besides, there are recent publications showing the existence of the Ebf1+ invasive fibroblast, not sure if the authors have detected this population of mesenchymal cells or not.

Response: We agree that mesenchyme is a crucial compartment in terms of responses to radiation and fibrogenesis. Following the reviewer’s suggestion, we analyze our data in more details and found that pericytes and mesotheliocytes represent respectively less than 0.35% (i.e. from 0 to 36 cells per sample) and 0,9% (i.e. from 5 to 97 cells per sample) of the cells analyzed per sample. Unfortunately, such low numbers preclude any detailed analysis of these populations. In fact, enabling deeper analysis would require to enrich these cell types using FACS, followed by RNA-seq. Regarding the existence of an Ebf1+ invasive fibroblast subset, Ebf1 expression was predominantly detected in the Col14a1+ fibroblasts populations but its expression was not significantly modified upon radiation injury (Figure 2 for review only).

Figure 2 (for review only) : Ebf1 expression in fibroblasts after radiation injury. (A) RIL dataset contains three subsets of fibroblasts annotated as Fibroblasts_Col13a1, Fibroblasts_Col14a1 and Myofibroblasts (A). Ebf1 is exclusively expressed in the Fibroblasts_Col14a1 subset (B) but its expression remains largely unmodified after radiation injury (C).

In the methods, the authors didn’t mention the sex of the mouse models. The authors should follow the ‘Sex and Gender Equity in Research – SAGER – guidelines’ and include sex and gender considerations for studies involving vertebrate animals.

Response: To follow the SAGER guidelines, we included the sex of the mice analyzed in the method section.

Reviewer #2 (Remarks to the Author):

In this study, Sandra Curras-Alonso et al. performed scRNA-seq of dissociated lungs from non-irradiated mice and mice 1, 2, 3, 4 and 5 months after fibrogenic (17 Gy) and non-fibrogenic (10 Gy) doses of IR, which provided a whole organ single cell atlas spanning the evolution over time towards pulmonary fibrosis, from the early response and the inflammatory phase to the late tissue reaction of fibrotic process. Firstly, the data of single cell transcriptome atlas aim at studying radiation-induced pulmonary fibrosis is indispensable resource. Secondly, some experiments, such as smFISH, in addition to sequencing data, were included for validation. Thirdly, an interactive web-based interface is useful to study lung responses to irradiation. Overall, this is a valuable study, but the following points should be addressed.

Major concern:

1. Quality control is important for single cell atlas. The quality measures of scRNA-seq data were insufficient in this study. More quality control analyses of scRNA-seq are needed.

Response: We totally agree with the reviewer’s view on the importance of the quality controls in single cell RNAseq dataset. We had included in the Materials and Methods section a perhaps too brief description of quality control steps we take when analyzing the data, and we deeply apologize for this.

We have now detailed the different steps of the quality control : at the level of the whole object, we performed a first series of standard QC based on the number of genes detected per cells (nFeature), the number of transcripts in each cell (nCount) and the proportion of mitochondrial genes (percent.mt) per sample. In a second step, we sub-clustered each main cell types annotated and we filtered out the doublets as well as the low quality cells based on the number of transcripts detected. This second step was initially done for the different cell types analyzed thoroughly in the manuscript (i.e. AT2, Fibroblasts, Macrophages and Endothelial cells) but we have not applied the thorough QC to the entire dataset. We have now updated Figure 1, Supplementary figure 1 as well as the interactive web interface with the cleaner dataset. In addition, for an easy access to the QC, we included the plots showing the number of genes detected per cells (nFeature), the number of transcripts in each cell (nCount) and the proportion of mitochondrial genes (percent.mt) per sample directly in the web interface.

2. Some interesting molecules in transdifferentiation were identified in response to fibrogenic irradiation, it is suggested to construct the regulatory network further from the scRNA-seq.

Response: We thank the reviewer for this important point. In order to identify critical regulons implicated in fibrogenic response to radiation, we implemented the analytic tool SCENIC (Single-Cell rEgulatory Network Inference and Clustering), developed by Stein Aerts' lab (<https://scenic.aertslab.org/>), using our AT2 single cell data in. Interestingly, albeit not unexpectedly, we found that regulatory networks driven by interferon-related transcription factors such as Stat1, Stat3 and Irf7 are specifically activated at 4 and 5 months after a dose of 17 Gy. This result highlights once more the importance of Interferon signaling in AT2 phenotype after fibrogenic dose of radiation, as suggested in the literature. We included these data in supplementary figure 2 and modified the text accordingly.

3. Why not further analyze transdifferentiation by Monocle? Monocle orders individual cells according to progress through a biological process.

Response: To further analyze the transdifferentiation process, we followed the reviewer's suggestion and applied Monocle to AT2 and AT1 subset. The trajectory analysis highlighted a link between AT2 and AT1 (shown in supplementary figure 2-e) and, furthermore, pseudotime ordering using the transdifferentiation genes confirmed the transition from AT2 to AT1 (shown in supplementary figure 2-f). Altogether, results from the trajectory analysis supported the conclusion that a transdifferentiation program is activated in AT2 in the months following an exposure to a fibrogenic dose of 17 Gy. This new analysis have now been included in the supplementary figure 2.

4. It would be better if the website supports the scATAC-seq data, for the data of the single cell transcriptome and paired chromatin accessibility atlas are quite precious.

Response: scATAC-seq data combined with transcriptomic dataset are indeed precious and highly informative resources. To our knowledge, scATAC-seq dataset from mouse radiation induced lung injury are not yet available and unfortunately, we have not generated scATAC-seq ourselves. In the future, if such dataset becomes available, we will integrate them into the website, along with our single cell transcriptomic data.

Minor comments:

1. The website built by authors should include a webinterface for data manipulation.

Response: The website now allows plotting of the gene expression for specific genes of interest and the UMAP visualization can be labelled using different types of metadata (e.g. cell type, condition). Other types of visualizations are available such as violin plots, box plots as well as analysis of the proportions. More in-depth analyses would require more computational power than what is normally supported by classical website. Therefore, for a more personalized analysis, users are expected to download the original data from GEO.

2. Proper formatting must be used; symbols for genes are in italics, and symbols for proteins are in non-italicized font. Corrections must be reflected in the figures and tables as well as text.

Response: Changes have been made throughout the text and figures.

3. “scRNAseq” should be "scRNA-seq", if it is the abbreviation for single-cell RNA sequencing.

Response: Done as recommended by the reviewer.

4. Page 6 line 173, the description “1M after IR17Gy 30% of these genes...” , suggest to rewrite as “1m after 17 Gy irradiation, 30% of theses genes...”.

Response: Changes have been made in the text as recommended.

5. Page 7 line 221, “the first subset is characterized by the expression of Folr2, Ccl8 and Cd163 (called here IM_C1)”. The Supplementary Fig. 4a did not show the expression of the Folr2 and please add it.

Response: Folr2 expression has been added to Supplementary Fig. 4a.

6. Page 10 line 353, the accession nos. of data for GEO was missing.

Response: We have now included the GEO accession number.

Reviewer #1 (Remarks to the Author):

The authors have satisfactorily addressed my concerns. But I found I had questions about Fig. 6. After these are addressed, I recommend the manuscript for publication.

1. Cell-cell interactions between fibroblasts Col14a1 and myofibroblasts as sources and gCap cells within Endothelial cells as targets were further investigated. Could the authors explain why these cell types were chosen to be further investigated?

2. It looks like within mesenchymal cells and endothelial cells, there are significant interactions going on, such as aCap and gCap, myofibroblasts and Col13a1 and Col14a1 fibroblasts. Therefore, I wonder what are the ligand-receptor pairs within mesenchymal and endothelial subclusters. Whether the collagen pathway is more significant within the mesenchymal cells or during the outreach to the endothelial cells.

Reviewer #2 (Remarks to the Author):

The authors addressed all of my concerns and revised the manuscript appropriately. The database replenished is a valuable resource for the readers and potential users.

We thank the reviewer for the time and effort in reviewing our manuscript and address the questions in the paragraphs below.

Reviewer #1

The authors have satisfactorily addressed my concerns. But I found I had questions about Fig. 6. After these are addressed, I recommend the manuscript for publication.

1. Cell-cell interactions between fibroblasts Col14a1 and myofibroblasts as sources and gCap cells within Endothelial cells as targets were further investigated. Could the authors explain why these cell types were chosen to be further investigated?

In our cell-cell communication analysis, the fibroblasts Col14a1 and myofibroblasts were the two mesenchymal subsets showing an increased communication through pathways linked to fibrogenesis such as Collagen and Fn1/Fibronectin. We decided to focus our interest on these interactions, emerging after 3 months, as they may have implications in fibrosis development and not solely characterize the fibrotic state of the lung at 5 months post-irradiation.

2. It looks like within mesenchymal cells and endothelial cells, there are significant interactions going on, such as aCap and gCap, myofibroblasts and Col13a1 and Col14a1 fibroblasts. Therefore, I wonder what are the ligand-receptor pairs within mesenchymal and endothelial subclusters. Whether the collagen pathway is more significant within the mesenchymal cells or during the outreach to the endothelial cells.

To address the question, we checked the strength of the collagen pathway interactions within the mesenchymal as well as within the endothelial cells. As presented in the plot below, the strength of the collagen pathway (i.e. information flow from Cellchat) is more significant in the interactions between Fibroblasts_Col14a1/Myofibroblasts towards gCap/aCap than between the fibroblasts (i.e. Fibroblasts_Col14a1→Fibroblasts_Col13a1, Myofibroblasts→Fibroblasts_Col13a1) or between the endothelial subsets (gCap→aCap).